



# Climate reconstructions based on GDGT and pollen surface datasets from Mongolia and Siberia: calibrations and applicability to extremely cold-dry environments over the Late Holocene.

Lucas Dugerdil[1,2], Sébastien Joannin[2], Odile Peyron[2], Isabelle Jouffroy-Bapicot[3], Boris Vannière[3], Boldgiv Bazartseren[4], Julia Unkelbach[5], Hermann Behling[5], and Guillemette Ménot[1]

[1]Univ. Lyon, ENS de Lyon, Université Lyon 1, CNRS, UMR 5276 LGL-TPE, F-69364, Lyon, France
[2]Université de Montpellier, CNRS, IRD, EPHE, UMR 5554 ISEM, Montpellier, France
[3]Université Bourgogne Franche Comté, CNRS UMR 6249 Laboratoire Chrono-environnement, F-25030, Besançon, France
[4]Ecology Group, Department of Biology, School of Arts and Sciences, National University of Mongolia, Ulaanbaatar 14201, Mongolia
[5]Department of Palynology and Climate Dynamics, Albrecht-von-Haller-Institute for Plant Sciences, University of Goettingen, Germany

**Correspondence:** Lucas Dugerdil, lucas.dugerdil@ens-lyon.fr

**Abstract.** Our understanding of climate and vegetation changes throughout the Holocene is hampered by representativeness in sedimentary archives. Potential biases such as production and preservation of the markers are identified by comparing these proxies with modern environments. It is important to conduct multi-proxy studies and robust calibrations on each terrestrial biome. These calibrations use large data base dominated by forest samples. Therefore, including data from taiga and steppe

sites becomes mandatory to better calibrate arid environments. The Mongolian plateau, ranging from the Baikal basin to the Gobi desert, is especially characterized by low annual precipitation and continental annual air temperature. The characterization of the climate system of this area is crucial for the understanding of Holocene Monsoon Oscillations. This study focuses on the calibration of proxy-climate relationships for pollen and glycerol dialkyl glycerol tetraethers (GDGTs) by comparing large published Eurasian calibrations with a set of 49 new surface samples (moss polster, soil and mud from temporary dry pond).

These calibrations are then cross-validated by an independent dataset of top-core samples and applied to two Late Holocene paleosequences in the Altai mountains and the Qaidam basin. We show that: (1) preserved pollen assemblages are clearly imprinted on the extremities of the ecosystem range but mitigated and unclear on the ecotones; (2) for both proxies, inferred relationships depend on the geographical range covered by the calibration database as well as on the nature of samples; (3) even if local calibrations suffer from reduced amplitude of climatic parameter due to local homogeneity, they better reflect

actual climate than the global ones by reducing the limits for saturation impact, (4) a bias in climatic reconstructions is induced by the over-parameterization of the models by addition of artificial correlation and (5) paleoclimate values reconstructed here are consistent with Mongolia-China Late Holocene climate trends, and validate the application of local calibrations for both pollen and GDGTs (closest fit to actual values and realistic paleoclimate amplitude). We encourage the application of this surface calibration method to reconstruct palaeoclimate and especially consolidate our understanding of the Holocene climate

and environment variations in Arid Central Asia.



# 1  Introduction

Since the understanding of the interactions between the palaeoclimate proxies, such as pollen or biomarker abundances, and General Circulation Model outputs became a major issue in future climate change modelling, resolving the issue of climate proxy calibration is crucial (Braconnot et al., 2012). Current climate changes in extremely cold environments (Masson-Delmotte, 2018), such as Mongolia and Siberia (Fig. 1), are amplified compared with other places around the world (Tian et al., 2014) and the drivers of the current degradation of Mongolian environments in diversity and biomass production still need to be understood. From a climatic point of view, Mongolia is on a junction between the westerlies which are driven by the North Atlantic Oscillation (NAO) and the East Asian Summer Monsoon which is linked to the El Niño-Southern Oscillations (ENSO) and the Inter-tropical Convergence Zone (ITCZ, An et al., 2008). The Mongolian plateau is a hinge area: the high altitude of the Altai range to the west and the Sayan range to the north-west of the country partially block both the westerlies arriving from the northern Atlantic ocean through the Siberian basin and the East Asian Summer Monsoon (EASM, Fig. 2, Chen et al., 2009). The Mongolian system is thus driven by a mix of the distant drag of these two main climatic cells. The understanding of the complex interaction of these cells is necessary and continuously improved by palaeoclimatic studies (**?**).

Lake sediment archives are commonly used to infer past variations of these climate and environmental systems associated with vegetation and human land use (Lehmkuhl et al., 2011; Felauer et al., 2012; Wang and Feng, 2013). Among the proxies available, pollen and geochemical biomarkers are used as past temperature indicators (Ter Braak and Juggins, 1993; Weijers et al., 2007b) and the combination of these proxies helps to polish lake sediment shift interpretations (Atahan et al., 2015; Watson et al., 2018; Martin et al., 2020; Kaufman et al., 2020). From decades the pollen signal is used to display shifts in vegetation composition and structure (Bennett and Willis, 2002) and allows quantitative reconstructions of climate parameters such as precipitation regime and temperatures (Birks et al., 2010; Ohlwein and Wahl, 2012; Wen et al., 2013; Cao et al., 2014; Marsicek et al., 2018). Since vegetation structure and pollen production are mainly influenced by climatic parameters (Zheng et al., 2008) in the absence of human influences, the paleo-pollen signal is very often interpreted as a response to the climate variations through time (Kröpelin et al., 2008; Wagner et al., 2019). Even if human activities influence pollen rain as well (Hjelle, 1997; Hellman et al., 2009a), these empirical observations of the pollen-climate relation leads to the development of semi-quantitative (Ma et al., 2008) and quantitative calibrations (Brewer et al., 2008; Salonen et al., 2019) of the signal. Different methods have been developed to reconstruct past climates: Probability Density Functions, Assemblages approaches, Transfer Functions (TF) and methods based on vegetation models (Guiot et al., 2000; Birks et al., 2010; Bartlein et al., 2011; Ohlwein and Wahl, 2012). More precisely, these methods are: the Inverse Modelling Method (IMM, Guiot et al., 2000), the Weighted Averaging Partial Least Squares regression (WAPLS, Ter Braak and Juggins, 1993; Ter Braak et al., 1993), the Artificial Neural Networks (ANN, Peyron et al., 1998), the Modern Analogue Technique (MAT, Overpeck et al., 1985; Guiot, 1990; Jackson and Williams, 2004), the Response Surface Technique (RST, Bartlein et al., 1986), Probability Density Functions (PDF, Kühl et al., 2002; Chevalier,





2019), Modified Mutual Climate Range Method (MMCRM, Klotz et al., 2003, 2004), Bayesian Hierarchical Models (BHM,

Ohlwein and Wahl, 2012), the Boosted Regression Trees (BRT, Salonen et al., 2014), etc. For northern Europe and despite some problems and pitfalls, Seppä et al. (2004) demonstrated that pollen-inferred climate reconstructions are generally consistent and agree well with other independent climatic reconstructions. This study encourages us to lead multi-proxy studies to refine climate reconstruction understanding, and especially in tricky and dry context such as Mongolian plateau (Rudaya et al., 2009).

Among new promising proxies and from the three last decades, biomarkers such as the glycerol dialkyl glycerol tetraethers (GDGTs) have provided new perspectives on continental temperature reconstructions (Naafs et al., 2017a, b). Among the GDGTs, we will focus on two major groups: the isoprenoid-GDGTs (isoGDGTs, Hopmans et al., 2000) and the branched-GDGTs (brGDGTs, Damsté et al., 2000; Weijers et al., 2007a, b; Dearing Crampton-Flood et al., 2020). BrGDGT assemblages reflect bacterial activity in rivers (De Jonge et al., 2014b), soil (De Jonge et al., 2014a) or lake water column (Dang et al., 2018).

Environmental drivers are linked to climate parameters (Weijers et al., 2007b), soil typology and vegetation cover (Davtian et al., 2016), which in turn imply land cover and land use. Accurate determinations of the relationships between brGDGT assemblages and climate still need some improvements (Naafs et al., 2018; Wang et al., 2019, 2020) and especially on local to regional scales and in extreme environments. It has been shown empirically (Weijers et al., 2004; Huguet et al., 2013) on cultures of pure strains (Salvador-Castel et al., 2019 in press) as well as on meso- and micro-cosm experiments (Chen et al.,

2018; Martínez-Sosa et al., 2020), that organisms adjust their membrane plasticity by the degree of methylation and cyclisation of the compounds. Moreover, some studies have focused on variations in the bacterial community structure (Xie et al., 2015), the bacterial group responses to environmental changes (Knappy et al., 2011) and the GDGT occurrences in different bacterial communities (Liu et al., 2012b) to determine the potential effects of community structure on GDGT relative abundances. To evaluate the provenance and the climatic information brGDGTs bear, several indexes have been proposed in the literature

(Supplementary Table S1). To monitor these changes, CBT and MBT indexes linked to environmental factors such as climate and soil parameters have been proposed (Weijers et al., 2007b; Huguet et al., 2013). In particular, the methylation degree, ratio of 5, 6 (De Jonge et al., 2013) and 7-methyl isomers (Ding et al., 2016) reacts to environment forcing : the 5-methyl correlates mainly with temperature (Naafs et al., 2017a), while 6 and 7-methyl seem to react to moisture and pH (Yang et al., 2015; Ding et al., 2016). More specific indexes have been proposed by De Jonge et al. (2014a) to limit the multi-correlation systems such

as $\mathrm{MBT}'_{5Me}$ which is independent of the pH and $\mathrm{CBT}_{5Me}$ which is more representative of the soil pH than the former version of the index. The statistical relevance of these indexes is a major issue in brGDGT calibration (Dearing Crampton-Flood et al., 2020). Some regional indexes for soil temperature such as $\mathrm{Index}_1$ (De Jonge et al., 2014a) and $\mathrm{Index}_2$ for Chinese soils (Wang et al., 2016) have been explored too, in a context of a strong local calibration demand (Ding et al., 2015; Yang et al., 2015). For moisture variations, the $\mathrm{R}_{i/b}$ index has been proposed as a reliable aridity proxy (Yang et al., 2014; Xie et al., 2012). It has

been shown that a linear relation exists between these GDGT indexes and some climatic features at large regional scales (in the wide Chinese biome range, from tropical forest to central arid plateau, for instance, Yang et al., 2014; Lei et al., 2016).



Since multi-proxy studies become more and more accurate in both temperature and precipitation reconstruction, local to regional calibrations have been proposed for dry areas such as the Arid Central Asian (ACA) area: pollen semi-quantitative climate reconstruction (Ma et al., 2008), pollen transfer functions (Herzschuh et al., 2003, 2004; Cao et al., 2014; Zheng et al., 2014), and brGDGT regression models (Sun et al., 2011; Yang et al., 2014; Ding et al., 2015; Wang et al., 2016; Thomas et al., 2017). Even if all of these studies focus on areas surrounding the EASM line (Fig. 2, Chen et al., 2010; Li et al., 2018), the understanding of the climate cells interaction and oscillation over time is still lacunary, and especially on the ACA upper edge. In this context, our study took place on the northernmost part of this climatic system (Haoran and Weihong, 2007). Moreover, we propose the first multi-proxy calibration exercise in ACA area based on pollen and brGDGT fractional abundances retrieved from modern samples (soil, moss litter, pond mud) in semi-arid to temperate conditions. The aim of this study is to take advantage of new, modern surface sample datasets in Siberia and Mongolia to propose an adapted calibration of pollen and bacterial biomarker proxies for cold and dry environments. For that purpose, local calibrations are compared with global calibrations to infer the influence of calibration scale and proxy types on derived climatic parameters. Our approach is summarized in the following steps:

1. Collection of a new set of modern surface samples for Mongolia with homogeneous characterisation of their bioclimate environment followed by pollen and GDGT pattern characterization.

2. Evaluation of the match between actual bioclimate environments and the associated pollen rain and biomarker assemblages based on mathematical criterion without eco-physiological considerations.

3. Creation of local Siberia-Mongolia climate calibrations for pollen and GDGTs and comparison of local and global calibrations on the Mongolian study case.

4. Posteriori validation of the inferred relationships between proxies and ecological likelihood based on the currently developed evidences of brGDGT and pollen rain ecological significance.

5. Discussion of the implications of the calibration mismatches in terms of climatic reconstructions in arid and cold environments.

6. Testing of the new calibrations (pollen and brGDGTs) through their application on two surrounding Late Holocene paleosequences.

## 2 Mongolian and Siberian Study Area

### 2.1 Coring, Sampling Area and Sample Types

The study area lies from $52°29'$N to $43°34'$N in latitude and from $101°00'$E to $107°06'$E in longitude (Fig. 1.A). The sample sites (n = 49) are listed in the Supplementary Tables S2 with a description of the sample type, the applied analyses, the coordinates and the associated ecosystem. For each site, the *Garmin eTreX10* was used to record GPS coordinates to five-meter





**Figure 1. A**: Topographic map of Mongolia (from ASTER data) with the location of surface samples and weather stations considered in the present study; **B**: Mean Annual Precipitations; **C**: Mean Annual Air Temperatures; **D**: Focus on the samples surrounding the Taatsiin Tsagaan Lake, Gobi desert; **E**: Focus on the samples along a valley in the Khentii range; **F**: Localisation of Khangai surface samples; **G**: Focus on the Baikal Lake transect. The Mongolian GIS Data is issued from a dataset ASTER (https://biosurvey.ku.edu/directory/nicholas-kotlinski), the meteorological dataset from WorldClim2 and infrastructures from public dataset (ALAGaC) (https://marine.rutgers.edu/ cfree/gis-data/mongolia-gis-data/)



accuracy. The surface samples were collected throughout Mongolia in 2016 following four transects *(n = 29)*: in the Khentii mountain range (*MMNT1* and *MMNT2*, Fig. 1.E), in the Orkhon valley (*MMNT3*), in the Gobi desert and the Gobi-Altai range
(*MMNT4*, Fig. 1.D). During the same field trip, a fifth transect has been done in the Sayan range from the Siberian Oblast of Irkoustsk, Russia (*MRUT1, n = 12*, on Fig. 1.G). A Khangai mountains field trip from spring 2009 enlarge this set of data with a sixth transect of surface samples (*MMNT5, n = 6*, on Fig. 1.F) and two lake coring from Ayrag Nuur (*MMNT5C12*) and Shargyl Nuur (*MMNT5C11*) both on Fig. 1.F. Both of these top-cores were added to the surface pollen database, while only the *MMNT5C12* core has been used as cross-value to check the accuracy of the brGDGTs climate models. This core has been
clipped into 62 samples of which the top-core has been replicated 6 times (samples *MMNT5C12-1 to MMNT5C12-6)*. Into the *MMNT5* transect, mud from two temporary dry ponds has been sampled. These surface muds are referred in following figures as *mud*. Into the other transects and depending on aridity and vegetation at each site, a soil or a moss polster was sampled. In figures, *soil* refers to the 3 to 5 first centimetres of the ground in dry ecosystems, while *moss* is a mix between soil, litter and a bryophyte (or cyperaceae) layer in wetter environments. Moss acts as a pollen trap recording a three to five-year mean
pollen signal (Räsänen et al., 2004). In drier areas, the soil surface samples have the same function, in spite of a lower pollen conservation and over-representation of some *taxa* (Lebreton et al., 2010). In parallel for the GDGT analysis and following the calibration approaches presented in De Jonge et al. (2014a), Davtian et al. (2016) and Naafs et al. (2017a, 2018) mud from temporary ponds, soil samples as well as soil part of moss litter were also used for actual GDGT analysis. To summarize, this study is based on 49 sites, 48 samples in the pollen dataset, 44 in the brGDGT dataset and 6 cross-validation samples to test
the brGDGT models. In terms of sample types, the dataset consists of 30 mosses, 15 soils, 2 pond muds and 2 top-cores.

To test the reliability of our modern calibrations, we have finally selected two paleosequences close to the Mongolian plateau used as test-benches of the calibrations. For the pollen analysis, the core D3L6 from Unkelbach et al. (2019) and located in the Mongolian Altai range is compared to the Xiangride section (XRD) used for brGDGT sequence from Sun et al. (2019),
sampled in the Chinese Qaidam Basin (Fig. 2). These two cores have recorded the paleoenvironmental changes of the Late Holocene period.

## 2.2 Vegetation and Biomes

The central part of the Mongolian plateau is characterized by a dry and cold flat desert with a 1220 m.a.s.l. median elevation (Fig. 1.A, Wesche et al., 2016) and is intersected in its northern part by the Khentii range and in its southern part by the
Gobi-Altai range aligned along a NW-SE direction. A wet and cold highland in the Khangai ranges culminates at 4000 m.a.s.l and a flatter and wetter Mongolian area, the Darkhad basin, is located in the north, close to the Russian border on the edge of the Siberian Sayan range. In the northernmost area, the geography is characterized by the Baikal lake basin at a lower altitude (around 600 m a.s.l, Fig. 1.G, Demske et al., 2005).

The distribution of vegetation and biomes follows a latitudinal belt organization: in the North the boreal forest presents a mosaic of light-taiga dominated by *Pinus sylvestris* mixed with birches (Demske et al., 2005). On the Mongolian plateau,





the dark-taiga dominated by larches (*Larix sibirica*) and Siberian pines (*Pinus sibirica*) also presents some spruces and fewer birches (*Betula* spp.). The Mongolian taiga is constrained to a region spanning from the Darkhad Basin to the Khentii range (Fig. 1.A). On the north face of the Khangai piedmont, the vegetation is dominated by a mosaic of forest-steppe ecosystems:

the steppe is dominated by the *Artemisia* spp. associated with Poaceae, Amaranthaceae, Liliaceae, Fabaceae and Apiaceae. On these open-lands there are some patches of taiga forest, following roughly the broadside and the northern face of the crest letting on to the grasslands in the valley. The two last vegetation layers in Mongolia through the elevation gradient is an alpine meadow dominated by Cyperaceae and Poaceae with a huge floristic biodiversity and an alpine shrubland with pioneer vegetation on the summits. On the southern slope of the range, the ecotone between the steppe and the desert vegetation extends

hundreds of kilometers from the northern part of the Gobi desert (with water supplied by the Gobi lake area in between) to the Gobi-Altai range in the south (Klinge and Sauer, 2019). In the southernmost part of the country, the warm and dry climate conditions favour desert vegetation dominated by Amaranthaceae, Nitrariaceae and Zygophyllaceae. The vegetation cover is lower than 25% and is mainly composed of short herbs, succulent plants and a few crawling shrubs.

## 2.3    Bioclimate Systems

In the central steppe-forest biome, the vegetation is marked by an ecotone with short grassland controlled by grazing in the valley and larches on the slopes. The forest is gathered in patches constituting between 10% and 20% of the total vegetation cover. There are also some patches of *Salix* and *Betula* riparian forests among the sub-alpine meadows on the upper part of the range. This vegetation is characteristic of the northern border of the Palaearctic steppe biome (Wesche et al., 2016). This

biome is characterised by a range of 800 to 1600 m a.s.l, a Mean Annual Air Temperature (MAAT, Fig. 1.C) between -2 and 2°C and a Mean Annual Precipitations (MAP, Fig. 1.B) from 180 to 400 $\mathrm{mm.yr^{-1}}$ (Wesche et al. (2016) based on Hijmans et al., 2005). In Mongolia, even if the MAP are very low ($\mathrm{MAP_{Mongolia}} \in [50; 500]\mathrm{mm.yr^{-1}}$), the major part of the water available for plants is delivered during late spring and early summer, in contrast to Mediterranean and European steppes (Bone et al., 2015; Wesche et al., 2016). These seasons are the optimal plant growth periods. Mongolian summer precipitations are

controlled by the East Asian Summer Monsoon system (EASM) instead of the Westerlies' winter precipitation stocked onto the Sayan and Altai range (Fig. 2; An et al., 2008).

## 3    Methods

### 3.1    Pollen Analysis, Modern Pollen Datasets and Transfer Functions

Different chemical processes were performed on the samples: bryophytic part of the moss samples were deflocculated by $KOH$ and filtered by $250\mu m$ and $10\mu m$ sieves to eliminate the vegetation pieces and the clay particles. Then, acetolysis was performed to destroy biological cells and highlight the pollen grains. For the soil and pond mud samples, 2 steps of $HCl$ and





$HF$ acid attacks were added to the previous protocol to remove all the carbonate and silicate components. All the residuals were finally concentrated in glycerol and mounted between slide and lamella. The pollen counts were carried out with a *Leica*

*DM1000 LED* microscope on a 40× magnification lens. The total pollen count size was determined by the asymptotic behaviour of the rarefaction curve. This diagram was plotted during the pollen count using *PolSais 2.0*, software developed in *Python 2.7* for this study. The rarefaction curve was fitted with a logarithmic regression analysis. The counter was suspended whenever the regression curve reached a flatter step (Birks et al., 1992). A threshold for the local derivation at $dx/dy = 0.05$ was set. For each sample, the total pollen count is usually around $n \in [350; 500]$ grains for steppe or forest and $n \in [250; 300]$

for drier environment such as desert and desert-steppe.

Among all of the pollen-inferred climate methods, the MAT and the WAPLS were applied in this study on 4 different modern pollen datasets, and on the D3L6 fossil pollen sequence to test the accuracy of these calibrations (Unkelbach et al., 2019, Figs. 1.A and 2). The MAT consists of the selection of a limited number of analogue surface pollen assemblages with their associated

climatic values (Guiot, 1990); while the WAPLS uses a Weighted Average correlation method on a limited number of Principal Components connecting the surface pollen fractional abundance to the climate parameters associated (Ter Braak and Juggins, 1993; Ter Braak et al., 1993). The first dataset, called New Mongolian-Siberian Database (NMSDB), is composed of pollen surface samples analysed in this study ($N = 49$, Fig. 2). The second one is the same subset aggregated to the larger Eurasian Pollen Dataset (EAPDB) compiled by Peyron et al. (2013, 2017). From this dataset of 3191 pollen sample sites, a pollen–

*Plant Functional Type* method was applied to determine the biome for each sample according to the actual pollen rain (Fig. 2, Prentice et al., 1996; Peyron et al., 1998). Then, only the Cold Steppe (COST) dominant samples were extracted from the main dataset and aggregated with the NMSDB to produce the COSTDB ($N = 430$ sites, figured by lozenge in Fig. 2). Finally, a scale-intermediate dataset of samples located within the Mongolian border merged with the Mongolian New dataset is presented as MDB ($N = 151$ sites). The relation between each taxon and climate parameter was checked and then the MAT

and WAPLS methods were applied with the *Rioja* package from the R environment (Juggins and Juggins, 2019).

### 3.2 SIG Bioclimatic Data

Because Mongolia and Siberia have relatively few weather stations (Fig. 1.A), climate parameters were extracted with R from the interpolated climatic database *WorldClim2* (Fick and Hijmans, 2017). We used Mean Annual Precipitation (MAP, Fig. 1.B), Mean Annual Air Temperature (MAAT, Fig. 1.C), as well as temperatures and precipitations for spring, summer and winter

($T_{spr}$, $P_{spr}$, $T_{sum}$, $P_{sum}$, $T_{win}$ and $P_{win}$), Mean Temperature of the Coldest Month (MTCO) and the Mean Temperature of the Warmest Month (MTWA) in this study to characterize the actual climate. Because the Mongolian plateau is not rich in weather stations, the *WorldClim2* database suffers of interpolation errors. The surface sites presenting inconsistent climate parameters (MAP < 0 or MAP < Season Precipitation) were removed from the global database. The elevation data and the topographic map originate from the *ASTER* imagery (Fig. 1.A). The biome type for each site derives from the *LandCover* database (Olson

et al., 2001), classification and field trip observations.





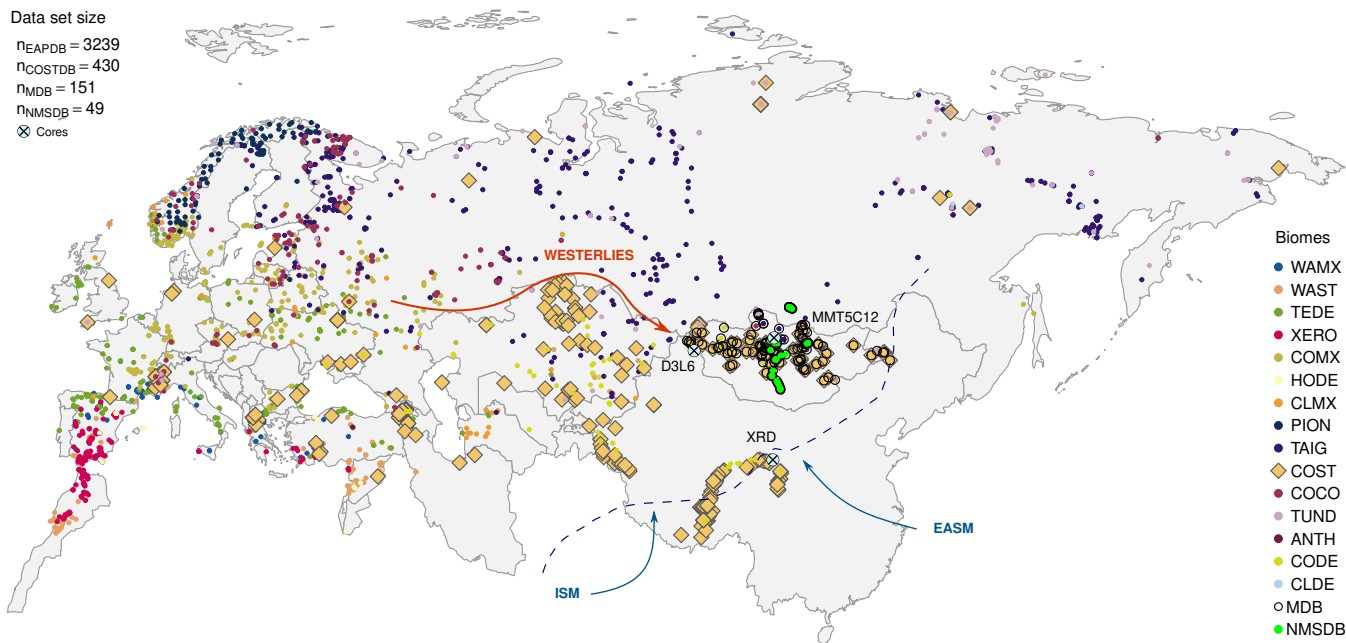

**Figure 2.** Eurasian map of all the pollen surface samples included in the database. The color code refers to the biome pollen inferred for each site. The biomes are **WAMX**, Warm Mixed Forest; **WAST** Warm Steppe; **TEDE**, Temperate Deciduous Forest; **XERO**, Xerophytic Shrubland; **COMX**, Cool Mixed Forest; **HODE**, Hot Desert; **CLMX** Cold Mixed Forest; **PION** Pioneer forest; **TAIG**, Taiga forest; **COST**, Cold Steppe; **COCO**, Cold Conifer Forest; **TUND**, Tundra; **ANTH**, Anthropic environment; **CLDE**, Cold Deciduous Forest; **CODE**, Cold Desert. The thickest points underline the COST samples selected for this study to operate the transfer function method among the whole COST sites (figured by lozenge on map). The arrows indicate the main climatic system driving the Mongolian climate: in orange the Westerlies arriving from the North Atlantic ocean and in blue the Indian Summer Monsoon (ISM) and the East Asian Summer Monsoon (EASM). The dashed line represents the EASM limit following Chen et al. (2010), (Li et al., 2018) and Haoran and Weihong (2007) for the northernmost part of the boundary. The light blue crossed circles localize the 3 cores used as test-benches for the calibrations (top-core and paleo-sequences).

### 3.3 GDGT Analysis

For consistency with the sampling process and the modelling methodologies developed for pollen analysis; soil part of the moss polsters, soil samples and pond mud were treated for GDGT analysis. After freeze drying, about 0.6 grams of material were sub-sampled. The Total Lipid Extract (TLE) was microwave extracted (*MARS 6 CEM*) with dichloromethane (DCM):MeOH

(3:1) and filtered on empty SPE cartridges. The extraction step was processed twice. Following Huguet et al. (2006), $C_{46}$ GDGT (GDGT with two glycerol head groups linked by $C_{20}$ alkyl chain and two $C_{10}$ alkyl chains) was added as internal standard for GDGT quantification. Then, apolar and polar fractions were separated on an alumina SPE cartridge using hexane: DCM (1:1) and DCM/MeOH (1:1), respectively. Analyses were performed in hexane: iso-propanol (99.8:0.2) by High Performance Liquid Chromatography Mass Spectrometery (HPLC-APCI-MS, Agilent 1200) proceeded in LGLTPE-ENS de Lyon,





Lyon, following Hopmans et al. (2016) and Davtian et al. (2018).

Each compound was identified and manually integrated according to its m/z and relative retention time (Liu et al., 2012a; De Jonge et al., 2014a). Statistical treatments on isoGDGT (Fig. 4.A) and brGDGT (Fig. 4.B) abundances were treated following two methods presented in Deng et al. (2016), Wang et al. (2016) and Yang et al. (2019): compounds were gathered by
chemical structures as cycles (CBT) or methyl groups (MBT, De Jonge et al., 2014a). brGDGTs were expressed as fractional abundance $[x_i]$ (Fig. 4.B, Sinninghe Damsté, 2016), as follows:

$$f[x]_i = \frac{n_i}{\sum\limits_{j=1}^{N_{brGDGT}} x_j} \tag{1}$$

To infer temperatures from brGDGT abundances, two types of model were applied: linear relationships between temperature and MBT–CBT indexes, and Multiple Regression (mr) models between one climate parameter and a proportion of multiple
brGDGT fractional abundances. For the simple linear regression model, a correlation matrix between climate parameters and indexes was calculated using the *corrplot* Rcran library. For mr models, we developed in the R environment a *Stepwise Selection Model* (SSM, Yang et al., 2014) to determine the best fitting model connecting climate parameters with brGDGTs fractional abundances. Then we gathered some of the climate–GDGT linear relations established in previous studies (De Jonge et al., 2014a; Naafs et al., 2017a, b, 2018; Sinninghe Damsté, 2016; Yang et al., 2014, 2019) focusing on a single climatic
parameter, MAAT (Supplementary Table S1). These models were clustered into 3 categories, by sample types (mosses, soils or pond muds), geographical area (regional or worldwide scale) and the statistical model (MBT-CBT based of multiple regression models). According to the type of environment from which the samples originated, there was peat, soil and lake-inferred modelling. All these models were applied to the Siberian–Mongolian surface samples, compared with the actual MAAT value at each site and applied to the brGDGT XRD section (Fig. 2, Sun et al., 2019).

## 3.4  Statistical Analyses

GDGTs and pollen data were analyzed with a Principal Components Analysis (PCA) to determine the axes explaining the variance within the samples. The biotic values (pollen and GDGTs) were also compared to abiotic parameters (climate, elevation, location and soil features) by the way of a Redundancy Analysis (RDA). The regression models were run with the $p-value < 0.05$ (model relevance), the $R^2$ (correlation level between the variables), the Root Mean Square Deviation (RMSE,
error on parameter reconstruction) and Akaike's information criterion (AIC, effect of over-parameterization on multiple regression models ; Arnold, 2010; Symonds and Moussalli, 2011). A cross-validation test was performed for all the brGDGT calibrations (from this study and from the literature) using an independent set of six lacustrine samples from the lake *MMNT5C12*. Statistical analyses were performed with the *Rcran* project, using the *ade4* package (Dray and Dufour, 2007) for multivariate analysis. All the plots were made with the *ggplot2* package (Wickham, 2016) or the *Rioja* package (Juggins and Juggins, 2019)
for the stratigraphic plot and the pollen clustering using the CONISS analysis method (Grimm, 1987).





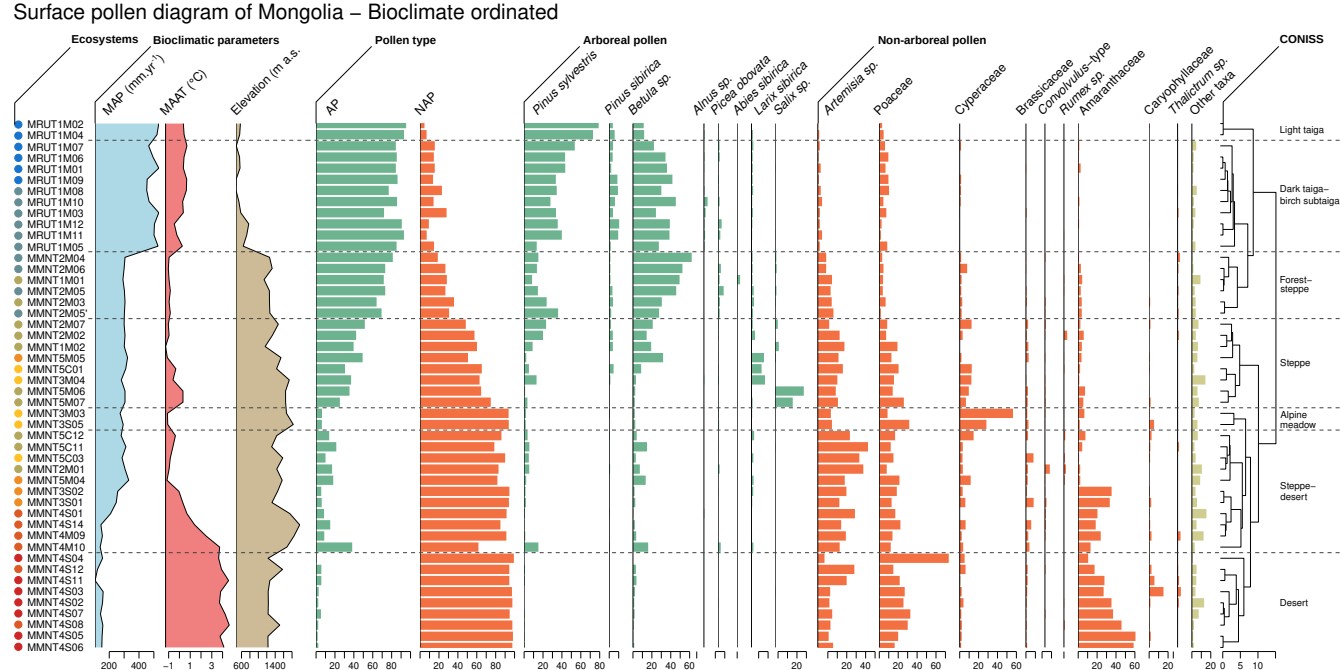

**Figure 3.** Simplified surface pollen diagram, bio-climatically sorted, of the Siberian–Mongolian transect. The pollen taxa are expressed in %TP. The Ecosystem Units were determined with a CONISS analysis. The left hand coloured dots represent the ecosystem for each sample from dark taiga (deep blue), light taiga, steppe-forest, alpine meadow, steppe, steppe-desert and desert (deep red). The color scale is presented in Fig. 5. The MAP and MAAT are extracted from Fick and Hijmans (2017).

## 4 Results

### 4.1 Pollen, Climate and Ecosystems Relations

#### 4.1.1 Modern Pollen Rain and Vegetation Representation

The pollen rain (Fig. 3) is dominated by six main pollen taxa: *Pinus sylvestris*, *Betula* spp., *Artemisia* spp., Poaceae, Cyper-
aceae and Amaranthaceae. The pollen diagram, sorted by bio-climate from the wet and relatively warm Siberian basin on the upper part to the dry-warm Gobi desert on the bottom, presents a net AP decrease from 85% to 5%. 34.26 % of the variance is explained by $PC_1$ extending from positive values associated with NAP (Amaranthaceae, Poaceae and *Artemisia* spp.) to negative values associated with AP (*Pinus* undet., *Betula* spp. *Picea obovata* and *Larix sibirica*, on Fig. 5.C). This trend shows the transition between ecosystems, marked by the seven main *CONISS* clusters (Fig. 3) and $PC_1$ and $PC_2$ variations (Fig. 5.C).
Below are the over-representative main *taxa* for each of the Siberian–Mongolian transect ecosystem:

1. **Light taiga** dominated by *Pinus sylvestris* ($> 70\%$), *Pinus sibirica* and very low NAP ($< 5\%$).





2. **Mixed dark taiga–birches sub-taiga** with an assemblage of *Larix sibirica*, *Picea obovata*, *Pinus sylvestris* and *P. sibirica*.

3. **Forest-steppe** ecotone same AP assemblages that the dark taiga ecosystem with 20% of *Artemisia* spp., plus occurence of Poaceae, Cyperaceae, *Thalictrum* spp. and *Convolvulus* spp.

4. **Steppe** still dominated by *Artemisia* spp. (30%) and rising Poaceae (25%), Brassicaceae

5. **Alpine meadow** overpowered by Cyperaceae up to 50 %, Poaceae, Brassicaceae, Amaranthaceae and *Convolvulus* spp.

6. **Steppe-desert** ecotone highlighting by the transition between Amaranthaceae–Caryophyllaceae community and Poaceae–*Artemisia* spp. assemblages.

7. **Desert** dominated by Amaranthaceae (from 25 % to 65 %) and by rare pollen-type Caryophyllaceae, *Thalictrum* spp., *Nitraria* spp. and *Tribulus* spp.

### 4.1.2 Pollen – Climate Interaction

The pollen rain trends follow similar variations than bio-climate parameters in MAP, MAAT and elevation (Fig. 3). Highest AP values are correlated to large MAP (up to 500 mm.yr$^{-1}$) and relatively high MAAT (around 1 °C), in the low range Siberian basin. Then the transition between AP and NAP dominance is marked by decreases in both MAAT (–1 °C) and MAP (300 mm.yr$^{-1}$) connected to the high-elevation Khangai range. Finally, the dominance of NAP in the Gobi desert area is linked to very arid conditions (MAP < 150 mm.yr$^{-1}$) and relatively warm MAAT (up to 4 °C). The correlation between the taxa themselves and climate parameters is $R^2 = 0.38$ (RDA, Fig. 5.D). Rise in MAAT is associated with that of Amaranthaceae, Poaceae, *Sedum*-type and Caryophyllaceae percentages. On the contrary, decrease in MAAT is associated with a rise in the AP and Cyperaceae, *Artemisia* spp. and Brassicaceae percentages. MAP, fairly related to $RDA_1$, rises with AP and decreases with NAP (Fig. 5.D). Finally, the elevation gradient favors *Artemisia* spp. and Cyperaceae for NAP and *Salix* spp. and *Larix sibirica* for AP (Fig. 5.D).

### 4.1.3 Pollen-inferred Climate Reconstructions: MAT and WAPLS Results

To reconstruct climate parameters from pollen data, MAT and WAPLS methods were applied on the four scales, modern pollen datasets and the ten climate parameters (Table 1). All these methods can be run with $n \in [1; 10]$ parameters: the number of analogues for MAT and the number of components for WAPLS. The best transfer functions among all of them were selected by the following approach: in a first step, for each climate parameter the methods fitting with the higher $R^2$ and the lower RMSE were selected. Then, in case the highest $R^2$ and the lowest RMSE were not applied for the same number of analogues or components, we chose the method presenting the lower number of parameters. Despite the small number of parameters relative to the number of observations, the method fits well (Arnold, 2010, Table 1). MAT method gives better $R^2$ in bigger DB than in smaller ones. Fitting increases with the diversity and the size of DB, since MAT is looking for the closest value between



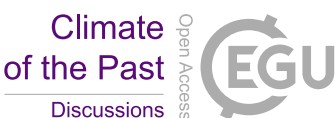

**Table 1.** Statistical results of the MAT and WAPLS methods applied to four surface pollen datasets and ten climate parameters[a].

| Database | Climate parameter | WAPLS | | | | | MAT | | | | |
|---|---|---|---|---|---|---|---|---|---|---|---|
| | | k best $R^{2(b)}$ | k best $RMSE^{(b)}$ | k selected[c] | $R^2$ selected[c] | RMSE selected[c] | k best $R^{2(b)}$ | k best $RMSE^{(b)}$ | k selected[c] | $R^2$ selected[c] | RMSE selected[c] |
| NMSDB | MAAT | 2 | 2 | 2 | 0.65 | 1.18 | **3** | 13 | 3 | 0.58 | 1.45 |
| | MTWAQ | 2 | 2 | 2 | 0.62 | 1.69 | **2** | 9 | 2 | 0.68 | 1.8 |
| | Tspr | 2 | 2 | 2 | 0.71 | 1.26 | **2** | 13 | 2 | 0.63 | 1.63 |
| | MAP | 2 | **1** | 1 | 0.79 | 61.22 | **2** | 6 | 2 | 0.88 | 55.73 |
| | Pspr | 2 | **1** | 1 | 0.67 | 11.39 | **2** | 4 | 2 | 0.89 | 8.28 |
| | Psum | 2 | 2 | 2 | 0.8 | 34.75 | **2** | 10 | 2 | 0.82 | 38.49 |
| MDB | MAAT | 2 | **1** | 1 | 0.35 | 1.9 | **5** | 8 | 5 | 0.6 | 1.65 |
| | MTWA | 2 | **1** | 1 | 0.24 | 2.14 | **5** | 9 | 5 | 0.53 | 1.84 |
| | MTCO | 2 | **1** | 1 | 0.27 | 2.75 | **5** | 7 | 5 | 0.66 | 2.05 |
| | MAP | 3 | **1** | 1 | 0.23 | 95.05 | **9** | 11 | 9 | 0.38 | 88.73 |
| | Psum | 1 | 1 | 1 | 0.47 | 47.02 | **8** | 12 | 8 | 0.54 | 46.17 |
| COSTDB | MAAT | 2 | 2 | 2 | 0.54 | 4.09 | **7** | 10 | 7 | 0.73 | 3.34 |
| | MTWA | 3 | **2** | 2 | 0.48 | 3.55 | **8** | 10 | 8 | 0.67 | 3.01 |
| | MTCO | 2 | 2 | 2 | 0.56 | 6.34 | **6** | 9 | 6 | 0.77 | 4.86 |
| | MAP | 4 | **2** | 2 | 0.55 | 224.43 | **6** | 9 | 6 | 0.77 | 161.86 |
| | Psum | 3 | **2** | 2 | 0.34 | 70.89 | **5** | 9 | 5 | 0.65 | 55.08 |
| EAPDB | MAAT | 3 | 3 | 3 | 0.72 | 4.08 | **5** | 8 | 5 | 0.88 | 2.9 |
| | MTWA | 3 | 3 | 3 | 0.55 | 3.31 | **5** | 9 | 5 | 0.79 | 2.5 |
| | MTCO | 3 | 3 | 3 | 0.72 | 6.49 | **4** | 8 | 4 | 0.89 | 4.46 |
| | MAP | 3 | 3 | 3 | 0.43 | 239.6 | **4** | 10 | 4 | 0.74 | 181.21 |
| | Psum | 2 | 2 | 2 | 0.52 | 62.33 | **4** | 8 | 4 | 0.8 | 44.66 |

[a] Only the five better fitting regression models for each climate parameter are shown .

[b] Corresponding to the number of parameters used in the model inferring the best $R^2$ and $RMSE$.

[c] Number of parameters, $R^2$ and $RMSE$ of the finally selected model.





climate and pollen abundance. By contrast, WAPLS fits better on the local scale and especially with a smaller number of sites. In this case, the pull of data is largest and the variance is largest (Ter Braak and Juggins, 1993). WAPLS also leads to better value of RMSE than $R^2$, in contrast to MAT. For temperature, pollen fits better with $T_{spr}$ or MTWA in Mongolia. Temperatures of the warmest months indeed control both vegetation extension and pollen production (Ge et al., 2017; Li et al., 2011) and especially in very cold areas such as Mongolia. For precipitation, the significant season is the one associated with the Summer Monsoon System in Mongolia (Wesche et al., 2016). Almost all the Mongolian precipitation falls during the spring and the summer (Wang et al., 2010), and the amount of precipitation controls, among other parameters, the openness of the landscape in Mongolia (Klinge and Sauer, 2019). To simplify the confrontation of the diverse models, the MAAT and MAP were isolated from the rest of the climate parameters. Even if these two climate parameters are not the best fitting pollen methods, they are the easiest to interpret and are comparable with the GDGT regression models commonly based on MAAT and MAP. In any case, these models are mitigated by the spatial autocorrelation affecting any models made on ecological database (Legendre, 1993) and especially the MAT method more than the WAPLS (Telford and Birks, 2005, 2011).

## 4.2 GDGT – Climate Calibration

### 4.2.1 GDGT Variance in the dataset

IsoGDGTs are dominated by $GDGT_0$ and crenarcheol (74,6% and 9.8% in relative abundances, respectively, in Fig. 4.A). These compounds were first attributed to lake water column production (Schouten et al., 2012), but they were also described in significant but lower proportions in soils (Coffinet et al., 2014). The variations of their fractional abundance in our soil and moss polster dataset are discrete and poorly linked to climate parameters (Fig. 4.A). IsoGDGT patterns in lake sediments do not really diverge from surface samples which leads to postulate that the *in-situ* production of isoGDGTs in shallow and temporary lakes like MMNT5C12 is reduced (Fig. 4.A). Then, it appears that the isoGDGT soil-produced are dominated by crenarcheol in accordance with studies on aridity impact (Zheng et al., 2015). However, no relationship exists between [crenarcheol] and MAP ($R^2 = 0.14, p - value > 0.005$). The putative regio-isomer reaction linked to MAP (Buckles et al., 2016) is not evidenced in NMSDB.

The average $[brGDGT]_{tot}$ concentrations differ depending on the sample type:

$$[brGDGT_{tot}]_{sed} = 25.8 \pm 8.1 \ ng.g_{sed}^{-1} \tag{2}$$

$$[brGDGT_{tot}]_{moss} = 23.2 \pm 26.8 \ ng.g_{moss}^{-1}$$

$$[brGDGT_{tot}]_{soil} = 0.3 \pm 0.14 \ ng.g_{soil}^{-1}$$

$$[brGDGT_{tot}]_{all} = 16.7 \pm 23.6 \ ng.g_{sample}^{-1}$$

brGDGT fractional abundances are consistent with each sample type: the predominant compounds are the $I_a$, $II'_a$, $II_a$ and $III_a$ (Fig. 4.B). These compounds explain dominantly the total variance (Fig. 5.A). Particularly, the $PC_1$ represents 22.77 % of the total variance and distinguishes two contrasted poles: the 5-methyl group (mostly with $PC_1 > -0.3$) associated with steppe-





forest and forest sites and the 6, 7-methyl group on the far negative $PC_1$ values associated with steppe and desert sites.


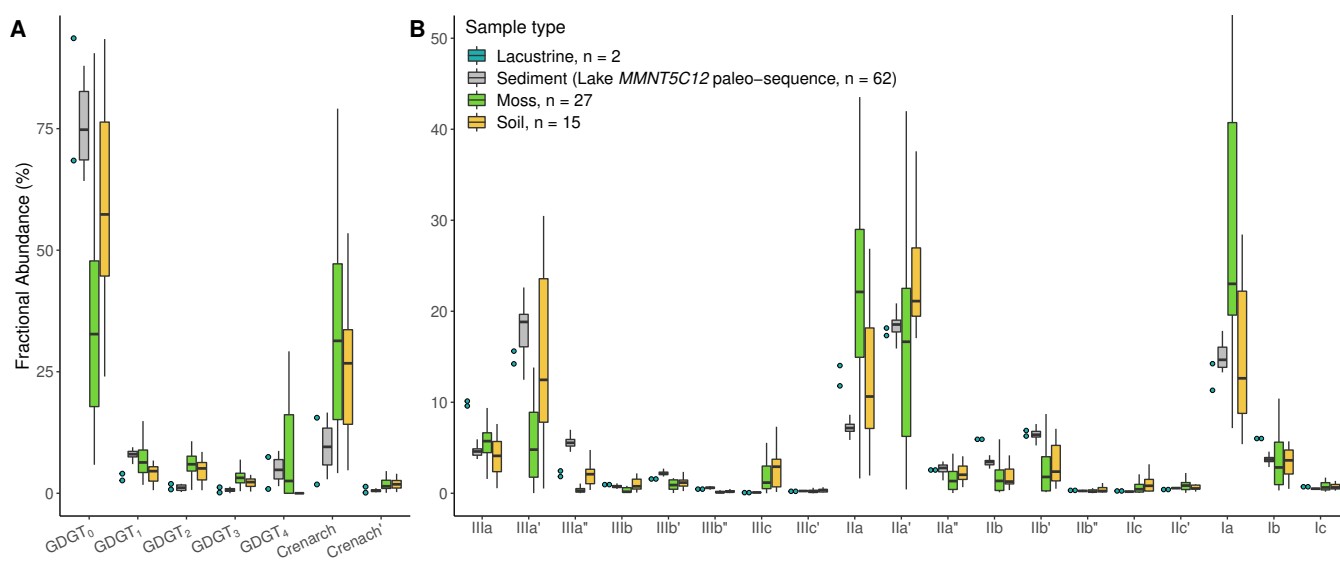

**Figure 4.** Fractional abundances of (**A**) isoGDGTs and (**B**) brGDGTs for moss polsters (green), soil surface samples (orange) mud from temporary dry pond (blue) and the full sequence of the Lake *MMNT5C12* as palaeo brGDGTs comparison (grey). The punctuation marks ' and " refer to 6 and 7 methyl, respectively.

The sediment samples from the lake MMNT5C12, used as past sequence comparison, are more homogeneous than the surface samples, especially when compared with the moss polsters that present a wide variability (Fig. 4.B). On this figure it appears that, globally, soil samples are more relevant analogues to sediments than moss polsters (especially the [IIIa'], [IIa] and [Ia] fractions in Fig. 4). This variability shows an influence of the sample type on brGDGT responses. On the other hand,

sample type also bears in first order climate and environment information, since soil and moss polsters originate mainly from steppe to desert environments and forest/alpine meadows, respectively.

### 4.2.2 Climate Influence on brGDGT Indexes

The brGDGT/climate RDA shows that the brGDGT variance is dominated by the MAP as first component (Fig. 5.B: $RDA_1 = 10.01\%$). The negative values show higher precipitations and uncyclized 5-Me GDGTs, such as $I_a$, $II_a$ and $III_a$, while the lower MAP

match with 6 or 7-Me GDGTs, such as $III'_a$, $II'_a$, $II''_a$ in accordance with De Jonge et al. (2014a). The $RDA_2$ is slightly more connected to MAAT opposed to elevation, also clustering the methyled and cyclized GDGTs to the higher MAAT. Such as in the pollen-climate response, the elevation is a second driving factor not to neglect. The correlation between relative abundance of methylated and cyclisated brGDGTs with climate parameters was not strong (Weijers et al., 2004; Huguet et al., 2013). All the MBT, MBT', $MBT'_{5Me}$ and CBT, CBT' , $CBT_{5Me}$ relations with climate parameters were tested and showed very





**Figure 5.** Multivariate statistics for the proxies clustered by ecosystems. **A**: Principal Components Analysis (PCA) and (**B**) Redundancy Analysis (RDA) for brGDGT fractional abundances; **C**: PCA and (**D**) RDA for pollen fractional abundances. The variance percentage explained is displayed on the axis label; the size of the dataset (n) and the RDA linear regression ($R^2$) are inserted in each plot area.





low correlation with $R^2 \in [0.1; 0.35]$ (Supplementary Fig. S3.B). Once the MBT (Supplementary Fig. S2.A) and the $\mathrm{MBT'_{5Me}}$

indexes (Fig. 6) compared with the world database (Yang et al., 2014; Naafs, 2017; Dearing Crampton-Flood et al., 2019) it

appears that the NMSDB set is consistent with known values instead of a while sample dispersion.

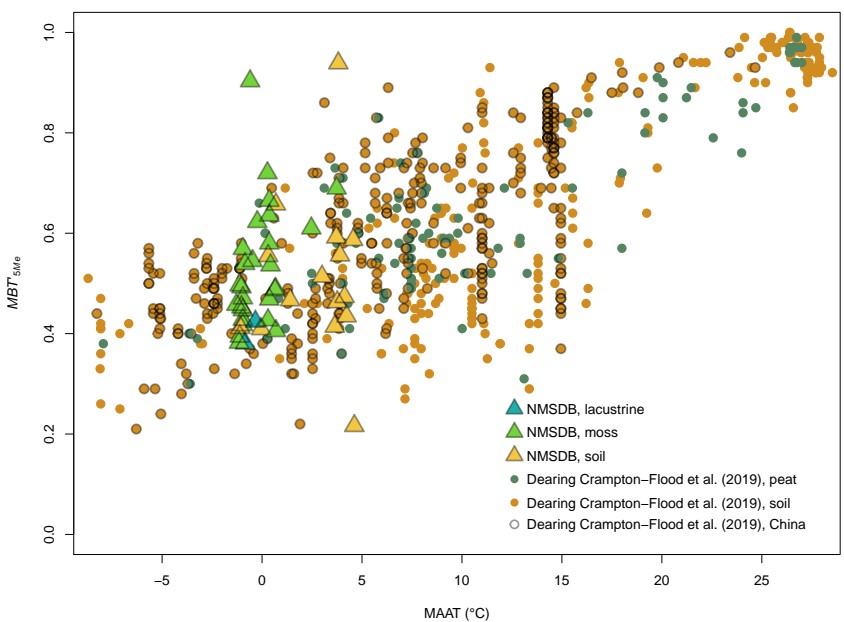

**Figure 6.** $\mathrm{MBT'_{5Me}}$–MAAT relation comparison between the NMSDB surface samples and the world peat and soil database Dearing Crampton-Flood et al. (2019).

### 4.2.3   Multi-regression Models

The Stepwise Selection Model for climate – brGDGT modelling was applied only on the 5 and 6 Methyls, because 7-methyled

brGDGTs show weak significance in the variance explanation (PCA, Fig. 5.A). The $\mathrm{N_{SSM}}$ different combinations of the 15

brGDGT compounds result in $\mathrm{N_{SSM}} = 2^{15} = 32768$ models possible for each climate parameter. Even the models including

some minor compounds ($[\mathrm{br}]_i < 5\%$) have been considered since, in the NMSDB, brGDGT fractional abundances are more

fairly distributed than in the global database, in which few compounds overlap the majority of the compound (Supplementary

Fig. S2.B). Indeed, the cumulative fractional abundance curve (Supplementary Fig. S2.C) is fairly faster to rich the asymptote

line for the world peat (blue curve) and the world soil (in brown) than in the NMSDB. The world peat database needs only three

brGDGTs to explain more than 85 % of the fractional abundance against more than ten compounds in the NDMSDB soils.

Then, the better fitting equations (with low RMSE and AIC and high R-squared) were selected for each number of parameters

(number of brGDGT issued in the linear regression) for both MAAT and MAP. Within the 15 models (one model for each





parameter addition), the 9 more contrasted ones were selected for discussion (Supplementary Table S3). The models with the

best statistical results were comprised of between 5 to 12 parameters and present a $R^2 \in [0.60; 0.76]$, a RMSE around 1.1 °C or 70 $mm.yr^{-1}$ and a $\mathrm{AIC_{MAAT}} \in [152.6; 166.2]$ or $\mathrm{AIC_{MAP}} \in [152.6; 166.2]$. The earlier a parameter is used in the mr models, the greater is its influence. For both $\mathrm{MAAT_{mr}}$ and $\mathrm{MAP_{mr}}$ models, $\mathrm{III_a}$, $\mathrm{III_a'}$, $\mathrm{III_b}$ and $\mathrm{III_b'}$ are the most relevant compounds for the climate reconstruction (Table 2) which is consistent with the PCA and RDA observations displayed (Fig. 5.A and B). Both the $\mathrm{MAAT_{mr}}$ models infer on a positive contribution of $\mathrm{III_a'}$ and a negative contribution of $\mathrm{III_a}$, which confirms

these models as eco-physiologically consistent with the RDA results. Moreover, except for $\mathrm{II_b'}$, all the compounds positively correlate with MAAT and negatively with MAP, in accordance with the MAP–MAAT anti-correlation. The $\Delta T$ values closest to 0 reveal the best fitting model on each point (Fig. 8, panel 1). Then, the box-plot (Fig. 8, panel 2) summarises the best fitting model at a regional scale.

**Table 2.** Statistical values and equations of the best brGDGT $\mathrm{MAAT_{mr}}$ and $\mathrm{MAP_{mr}}$ models.

| Model | Formula | $R^2$ | RMSE | AIC |
|---|---|---|---|---|
| $\mathrm{MAAT_{mr3}}$ | $= 0.6 \times 1 - 25.1 \times [\mathrm{IIIa}] + 12.3$ $\times [\mathrm{IIIa'}] + 7.2 \times [\mathrm{Ib}]$ | 0.57 | 1.3 | 156.4 |
| $\mathrm{MAAT_{mr5}}$ | $= 4.8 \times 1 - 38.5 \times [\mathrm{IIIa}] + 7.9 \times [\mathrm{IIIa'}]$ $-27.3 \times [\mathrm{IIIc}] - 3.3 \times [\mathrm{IIa'}] - 26.3 \times [\mathrm{IIb}]$ $+8.5 \times [\mathrm{IIb'}] - 5.6 \times [\mathrm{Ia}]$ | 0.66 | 1.1 | 153.9 |
| $\mathrm{MAP_{mr6}}$ | $= -511.3 \times 1 + 1205.9 \times [\mathrm{IIIa}] + 1387.2 \times [\mathrm{IIIb}]$ $+738.8 \times [\mathrm{IIa}] + 969.8 \times [\mathrm{IIa'}] + 1957.1 \times [\mathrm{IIb}]$ $+3006.3 \times [\mathrm{IIb'}] + 2406.4 \times [\mathrm{IIc}] + 2003.1 \times [\mathrm{IIc'}]$ $+1081.7 \times [\mathrm{Ia}] - 2406.3 \times [\mathrm{Ic}]$ | 0.73 | 72.5 | 525.8 |
| $\mathrm{MAP_{mr7}}$ | $= -502.6 \times 1 + 1359.5 \times [\mathrm{IIIa}] + 2462.5 \times [\mathrm{IIIb}]$ $-2178.3 \times [\mathrm{IIIb'}] + 657.7 \times [\mathrm{IIa}] + 986.8 \times [\mathrm{IIa'}]$ $+2440.5 \times [\mathrm{IIb}] + 3423.5 \times [\mathrm{IIb'}] + 2831.2 \times [\mathrm{IIc}]$ $+1967.2 \times [\mathrm{IIc'}] + 1150.2 \times [\mathrm{Ia}] - 955.6 \times [\mathrm{Ib}]$ $-2103.4 \times [\mathrm{Ic}]$ | 0.75 | 68.5 | 524.8 |



## 5 Discussion

### 5.1 Issues in Modelling Mongolian Extreme Bioclimate

#### 5.1.1 Appraisal Modelling in Arid Environments

According to Dirghangi et al. (2013) and Menges et al. (2014) ; the commonly used brGDGT indexes (MBT and CBT) are not relevant for arid areas with $\mathrm{MAP} < 500\mathrm{mm.yr^{-1}}$ because of the relation between low soil water content and soil brGDGT preservation and conservation interferes in the brGDGT / climate parameters (Dang et al., 2016). The MAAT models based on MBT and MBT' indexes provide colder reconstructions (Fig. 8.C2) as shown by De Jonge et al. (2014a), because arid soils favors 6-Methyl (by pH raising due to low weathering effect of the weak precipitation, Dregne, 1976; Haynes and Swift, 1989) and drives the MBT to decrease towards zero. This explains the colder $\mathrm{MAAT_{Ding}}$ and $\mathrm{MAAT_{MBT'_{DJ}}}$ reconstructed values compared to the modern ones. Moreover, the main issue in climate modelling in Mongolia is the narrow relationship between MAAT and MAP. Because of the climatic gradient from dry deserts in the southern latitudes to wet taiga forests in the northern ones, MAAT and MAP maps are strongly anti-correlated (Fig. 1 .B, C and Supplementary Fig. S1). If this correlation is not statistically determined on the range of the global database ($R^2 = 0.35$, $p < 0.005$), the impact is significant on the range of the Mongolian sites ($R^2 = 0.91$, $p < 0.005$). This correlation could be a bias resulting from the interpolation method of the *WorldClim2* database. In fact, there are very few weather stations (Fig. 1.A, Fick and Hijmans, 2017) and their distribution on the large Mongolian plateau is interrupted by mountain ranges. According to Fick and Hijmans (2017) the interpolation model used in the Central Arid Area (which includes our study area) gives a strong correlation ($\mathrm{R}^2 = 0.99$) and a little error ($\mathrm{RMSE} = 1.3°\mathrm{C}$) for MAAT and $\mathrm{R}^2 = 0.89$ and $\mathrm{RMSE} = 23\mathrm{mm.yr^{-1}}$ for MAP. Whenever the Siberian-Mongolian calibrations are used for palaeoclimatic reconstructions, the RMSE of the climate parameters has to be added to the RMSE model. Moreover, the relevance of the interpolation models suffers from the transition threshold made in Mongolia between the EASM and the Siberian Westerlies (Fig. 2, An et al., 2008) and reinforced by the topographic break (Fig. 1.A). Because the mr–GDGT models have been compiled with the group of Siberian sites which are out of the MAAT–MAP strong auto-correlation range (Supplementary Fig. S1), the reliability of the independence of the MAAT and MAP models seems to be guaranteed.

The topographic fence in Mongolia also affects the pollen and brGDGT distributions by it-self, as seen in both RDA analyses (Fig. 5.B and D) where elevation appears to be a main ecophysiological parameter. Elevation affects vegetation and pollen rain not just because of its influence on local MAAT and MAP but also because it drives other ecophysiological parameters such as $0_2$ concentration, wind intensity, slopes and creeping soils, snow cover and exposure (Stevens and Fox, 1991; Hilbig, 1995; Klinge et al., 2018). Elevation as one of the main brGDGT drivers could also be explained by the bacterial community responses to pH, moisture and soil compound variations along the altitude gradient (Laldinthar and Dkhar, 2015; Shen et al., 2013; Wang et al., 2015) and the vegetation shifts (Lin et al., 2015; Davtian et al., 2016; Liang et al., 2019).



### 5.1.2 Particularity of the Siberian–Mongolian climate system

Both GDGT and pollen calibrations show that the precipitation calibrations are more reliable than temperature ones (Tables 1, 2, Figs. 3, 7 and 8), reflecting that the Siberian-Mongolian system seems to be mainly controlled by precipitation. This dominance of precipitation could be due to seasonality. Even if the brGDGT production is considered to be mainly linked to

annual temperature means (Weijers et al., 2007a, b; Peterse et al., 2012), the high pressure Mongolian climate system (Zheng et al., 2004; An et al., 2008) favors a strong seasonal contrast: almost all the precipitation and the positive temperature values happen during the summer (Wesche et al., 2016). Consequently, for the NMSDB pollen transfer functions, the seasonal parameters such as MTWA, $T_{sum}$ and $P_{sum}$ better describe the pollen variability than MAAT and MAP climate parameters (better $R^2$ and RMSE in Table 1). While the opposite is found on EAPDB and COSTDB models, the calibration made on

large scale databases. The Mongolian permafrost persists half the year in the northern part of the country (Sharkhuu, 2003) and acts on vegetation cover and pollen production (Klinge et al., 2018). Furthermore, the effects of frozen soils on soil bacterial communities and GDGT production are thought to be important (Kusch et al., 2019). This seasonality leads to a quasi equivalence between MAP and $P_{Sum}$ (if $P_{win} \approx 0$ then $MAP \approx P_{sum}$) while MAAT is torn apart by the large $T_{Sum} - T_{win}$ contrast (because the MAAT is an average value and not a sum as for MAP). The mathematical consequences of the season-

ality on these two climate parameters are not the same. Finally, the MAP appears to be the most reliable climate parameter for Siberian-Mongolian climate studies according to the NMSDB sites (with $MAAT < 5°C$). Even if the brGDGTs seem to react to summer temperature (Wang et al., 2016; Kusch et al., 2019), the summer mr-models are not significantly improving the calibration compared to the $MAAT_{mr}$ ones. For instance, the best $Tsum_{mr}$ is selected by its AIC, $Tsum_{mr6}$ is inferred using 6 brGDGTs fractional abundance ( $R^2 = 0.63$ and $RMSE = 1.53°C$). This lack of seasonality effect, expected in such

cold areas, is consistent with temperate Chinese sites (brGDGT reconstructions, Lei et al., 2016).

### 5.1.3 Extreme Bioclimatic Condition Modelling Lead to a Better Global Climate Understanding

To reduce the signal/noise ratio, a wider diversity of sample sites should be added as initial inputs in the models. This raises the question of the availability of reliable samples in desert areas. The soil samples in the steppe to desert biomes are often very

dry and these over-oxic soil conditions are the worst for both pollen preservation (Li et al., 2005; Xu et al., 2009) and GDGT production (Dang et al., 2016). brGDGT concentrations in moss polsters and temporary dry pond muds are thus higher than in soils in our database (equation 2 and Fig. 4). The explanation of the signal difference between the three types of samples could also originate from the *in-situ* production of brGDGTs inside the moss predominant over the wind-derived particles brought to the moss net. As well, it seems that the pool of moss polster is associated with a similar trend than the worldwide peat samples

from Naafs et al. (2017b) (Supplementary Fig. S2.A and Fig. 6). Moreover, in the steppe or desert context of poor availability in archive sites, the edge clay samples or top-cores of shallow and temporary lakes could be a solution for palaeo-sequence studies. The two pond mud samples of the NMSDB are included within the soil-moss trend for all models (Fig. 5, Supplementary Fig. S2 and S3).Even if the brGDGT production and concentrations are different in soils than in lakes due to lake





*in-situ* production (Tierney and Russell, 2009; Buckles et al., 2014), this effect is function of the lake depth (Colcord et al.,
2015), consequently negligible for shallow lakes, and almost absent for lake edge samples as shown by Coffinet (2015) for
Lake Masoko in Tanzania.

The soils of the Gobi desert also have a high salinity level which is also a parameter of control on brGDGT fractional
abundances (Zang et al., 2018). This taphonomic bias (also climatically induced) could explain part of the histogram variance
of Fig. 4 related to the sample type as well as the shift of the soil–cluster from the regression line in the cross-value plot of
brGDGT MBT'/CBT models in Supplementary Fig. S3. Even if the impact of salinity on sporopollenin is not well understood,
salt properties may affects pollen conservation in soils (Reddy and Goss, 1971; Gul and Ahmad, 2006).

Finally, the saturation effect of the proxies when they reach the limits of their range of appliance is also to be taken into
considerations. Since both pollen and brGDGT signals are analysed in fractional abundance (i.e. % of the total count of con-
centration), these proxies evolve in a $[0;1]$ space. The saturation effect appears when extreme climatic conditions are reached
(Naafs et al., 2017a, b). For instance, in a tropical context, temperature values are too high to be linearly linked to fractional
abundances (Pérez-Angel et al., 2019). Considering pollen–climate relationships, the inferior limit of pollen percentage is crit-
ical: for the majority of pollen types, whenever MAAT or MAP reaches a very high or low threshold, the pollen fractional
abundance approaches zero (Fig. 9). These limit areas need to be closely investigated, which legitimises the local calibration
methods.

## 5.2   Statistical Tools for Best Model Selection

### 5.2.1   Over-Parameterization and Best Models Selection

Among the possible methods, statistical values help to select the most reliable ones for palaeoclimate reconstruction. However,
the correlation ($R^2$) and errors (RMSE) are not informative enough to discriminate between methods and to point to the
most suitable ones for palaeoclimate modelling. This is especially true for the multi-parameter methods (such as brGDGT
multi-regression models and pollen transfer functions). Indeed, the more input parameters in the method, the more accurate
it is (Tables 1, Supplementary Table S3 and Fig. 7.A and 7.B). All the regression models improve with parameter additions,
and especially the less fitting methods improve exponentially (lower limit of the $R^2$ area, Fig. 7.B). The best R-squared-
models for each parameter number (Fig. 7.A) correspond to the upper limit of the $R^2$ area (Fig. 7.B). This figure shows
that the $R^2$ vs. *parameter number* trend follows a logistic regression (both for $\text{MAAT}_{\text{mr}}$ and $\text{MAP}_{\text{mr}}$ models). However,
and especially for $\text{MAAT}_{\text{mr}}$ regression models, this logistic curve becomes asymptotic early, similar to the RMSE decrease.
The over-parametrization of the models has proven to produce artefacts in ecological modelling (Arnold, 2010; Symonds and
Moussalli, 2011). The issue is thus to identify the threshold in the parameter numbers selected. We used Akaike's Information
Criterion (AIC) to determine the better model without over-parameterization for brGDGT regression models: the lower the
AIC, the better the model (Supplementary Table S3 and Table 2). The trend of AIC versus the parameter number is however





more complex (Fig. 7.C). For $MAAT_{mr}$, the regression model becomes more accurate from one to five parameters rapidly, but then slowly decreases. The AIC curve takes an asymmetrical hollow shape around five parameters with a steeper slope on the left side (Fig. 7.A). The AIC values for $MAAT_{mr6}$ and $MAAT_{mr7}$ are almost identical (Fig. 7.A) . The $MAP_{mr6,7,8}$

have almost equivalent AIC values, while the AIC curve shapes differ for the other $MAP_{mr}$ models (asymmetrical hollow shape around five with a steeper slope on the left side, Fig. 7.A). To summarize, the most universal models are $MAAT_{mr5}$ and $MAP_{mr7}$ (Table 2) but the closed models are also valuable in some local contexts, and especially in similar dry-cold regions. We need to determine the cross-values of these models to select the appropriate ones for the Siberian-Mongolian context.

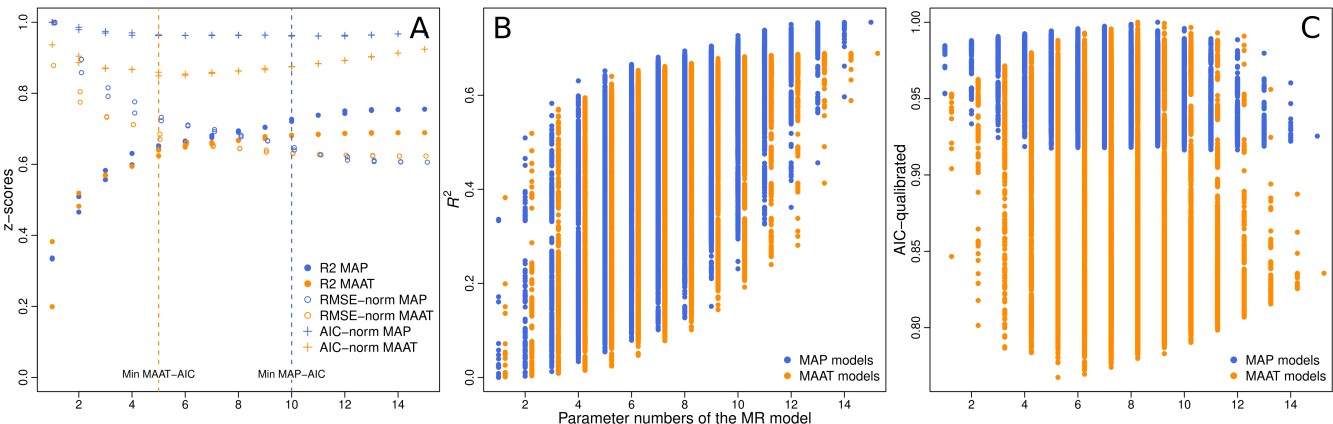

**Figure 7.** Statistical values plotted against the number of parameters of the different mr–GDGT models: the $R^2$, the RMSE normalized on the highest RMSE value and the AIC also normalized. **A**: selection of the two best multi-regression models for each number of parameters; **B**: combination of the $R^2$ (**B**) and the AIC (**C**) values for of all the mr models. The blue dots are for the MAP-models, orange dots for the MAAT one.

### 5.2.2 Assessment of the Calibration Feedback

The cross-values of the nine best $MAAT_{mr}$ regression models (Fig. 8.A1 and 8.A2) and the best $MAP_{mr}$ regression models (Fig. 8.B1 and 8.B2) were tested. The MAAT reconstructions provide different responses to the three main bio-climate areas (parcel A1): if they properly estimate temperatures in the Siberian Baikal basin, they overestimate and underestimate them for the center of the northern Mongolian mountains and the Gobi desert, respectively. For precipitation (parcel B1), all the $MAP_{mr}$ calculated with local to regional databases also misrepresent the extreme values: precipitation values are too high and too low

for the Gobi desert and the Baikal basin, respectively. To conclude, the wider the dataset extension, the more alleviated the extreme values.

Both on $MAAT_{mr}$ and $MAP_{mr}$ models, the 95 % interval shrinks with parameter addition, but the mean values do not necessarily get closer to the measured value of the climate parameter (the dashed line in Fig. *8.A2 and B2*). Therefore, if the





**Figure 8.** Validation of brGDGT-climate models on the study sites: reconstructed values for literature MAAT (**A**), NMSDB mr–MAAT (**B**) and MAP (**C**). Models are tested on the NMSDB sites (**1**) and the box-plot statistics (**2**) are provided. Sites are clustered in 4 groups: cross-value on the 6 first samples of the independent core MMNT5C12, Arkhangai; moss polsters from Mongolian steppe-forest; Gobi steppe-desert soil samples and moss polsters from Baikal basin. Values are plotted in anomaly.





tests on the AIC point toward the $\mathrm{MAAT}_{mr4}$ and the $\mathrm{MAP}_{mr7}$ regression models, the back-cross plots suggest the $\mathrm{MAAT}_{mr3}$
and the $\mathrm{MAP}_{mr6}$ regression models (Supplementary Table S3, coloured in blue and Table 2) provide the best estimates for
climate reconstruction in lacustrine archives ($\Delta \mathrm{MAP} = 0$ and best fitting temperature for the mean value of all samples, Fig.
8.B2 and Fig. 8.B1).

### 5.2.3   Global vs. Local Calibration

Whatever proxy is used, when reconstructing temperatures and precipitation from past records in a given location, there is
the issue of basing reconstructions on calibrations based on local or global datasets (among others, Tian et al., 2014; Cao
et al., 2014; Ghosh et al., 2017; Dearing Crampton-Flood et al., 2019). We tested both approaches on our datasets with a
cross-value run on the NMSDB-independent set of MMNT5C12 core samples. The global brGDGT - climate calibration ar-
tificially reaches higher R-squared than local ones due to the larger range of values of the involved climate parameters. Since
the world soil database in Naafs et al. (2017) covers a wide temperature range ($\mathrm{MAAT} \in [-5; 30]$), counter to the NMSDB
($\mathrm{MAAT} \in [0; 5]$), then its signal/noise ratio gets lower (Fig. 6). Despite the relatively lower R-squared of 0.62 scored by the
$\mathrm{MAAT}_{mr5}$ compared with world calibrations (Pearson et al., 2011; De Jonge et al., 2014a; Naafs et al., 2017a, b), the boxplots
for the all $\mathrm{MAAT}_{mr}$ calculated from the NSMDB are mostly centred on the MAAT average value with the shortest variance
spreading for all the sites (Fig. 8.C1 and 8.C2). These local calibrations fit best with the $\mathrm{MAAT}_{Ding}$ from Ding et al. (2015)
which is also a local calibration made on the Tibet-Qinghai plateau database. The global databases made on worldwide sites
(De Jonge et al., 2014b, a; Naafs et al., 2017a, b) provide $\mathrm{MAAT}_{model} > \mathrm{MAAT}_{real}$ and large standard deviation (SD). These
global calibration also attenuate the exterme MAAT values: the very cold Siberian basin / Mongolian plateau sites are recon-
structed with warmer temperatures up to +5 to + 10 °C, while the warm Gobi desert sites are down by up to -3 to - 5 °C.
On the other hand, the local calibrations performed on subtropical to tropical Chinese transects (Yang et al., 2014; Thomas
et al., 2017) have smaller SD but largely overestimate MAAT values due to the warmer conditions of the initial database sites.
In brief, the lake core sediment samples match the best to the modern MAAT and MAP value with the brGDGTs mr–models
which invite us to consider that these local brGDGT calibrations present a robust way to approach past climate.

Similarly, for pollen transfer functions, the geographic range of the surface samples on which the calibration relies is a rel-
evant parameter to take into account for the reliability of the paleoclimate reconstructions. The choice of the maximum value
of this geographic range has been discussed previously for vegetation modelling, for example, the Relevant Source Area of
Pollen (RSAP, Prentice, 1985; Hellman et al., 2009a, b; Bunting and Hjelle, 2010). For MAT and WAPLS regression models,
the same issue holds true. The responses of the eight over-represented *taxa* to climate parameters are different in the three
geographic ranges (NMSDB, COSTDB and EAPDB). The linear tendency allows for checking the main trends between taxa
distribution and climate parameters, despite the weak linear regressions ($p-value > 0.005$ and $R^2 < 0.4$, in Fig. 9). For the
majority of these *taxa*, the trend is the same, independent of the database size (*Larix* spp. and Cyperaceae percentages in-
creasing with weaker MAAT, or Amaranthaceae and *Pinus sylvestris* percentages increasing with higher MAAT). However,
due to the shift between pollen types and their associated vegetation (i.e. Poaceae-pollen signal similar for a wide diversity





**Figure 9.** Relationships between the eight major pollen *taxa* (%TP) and MAP ($\mathrm{mm.yr^{-1}}$, upper part of the facet plot) and MAAT (°C, lower part). The black line is the linear fitting for all samples (EAPDB), the orange for all the samples from steppe biome (COSTDB) and the blue only for the NSMDB samples presented in this article.




of Poaceae communities with very contrasted ecophysiological features), trends are controlled in some peculiar cases by the geographical clipping of the DB. Thus, Poaceae have a positive response to MAP on the global scale but not inside the Mongolian area. The human influence on pollen rain is also dependent on the biogeographical context, thus, *Artemisia* spp. is not considered as much human influenced in the Asian steppe environment (Liu et al., 2006) than in the European one (Brun, 2011).

Concerning transfer functions, WAPLS performs better for the local database than for the COST and EAP database (Table 1). On these subsets, the WAPLS RMSE and R-square values are even higher than for the MAT transfer function. The major difficulty resides in the reconstructions of precipitation. Even if the RMSE and $R^2$ values are higher for all models of MAP than MAAT, the influence of precipitation on vegetation cover is not well understood. In Mongolia it is clear that the precipitation controls the treeline in mountainous areas (Klinge and Sauer, 2019) and the global openness in the steppe - forest ecotone (Wesche et al., 2016) as well as human land-use (Tian et al., 2014), but the risk of autocorrelation between MAAT and MAP signals is important, even if the RMSE and $R^2$ values are higher for MAP regression models than for MAAT ones (Telford and Birks, 2009; Cao et al., 2014). Tangibly, for the two proxies, even if the global calibrations can operate on our study area, the local calibrations reach higher accuracy.

### 5.2.4 Test-bench of the Local Calibrations on Two Paleosequences

To test the reliability of our local calibrations, the pollen transfer function and the brGDGT mr-models have been applied on paleo-sequence. Because there is still no available core analyzed both for pollen and brGDGTs in ACA, the D3L6 core (pollen, Altai, Unkelbach et al., 2019) and the XRD section (Qaidan, GDGTs, Sun et al., 2019) are used. The actual values of the climatic parameters are first compared to the top-core reconstructed climatic parameters (Fig. 10, dashed lines). The amplitude of the variations through time has then to be assessed with regards to the expected regional ranges (Zheng et al., 2004). Finally, reconstructions on known short term climate events are tested for the last 5000 years. They are namely the Little Ice Age, Warm Medieval Period, Dark Ages Cold Period, Roman Warm Period and 4.2k year event (respectively LIA, WMP, DACP, RWP and 4.2 ; Zhang et al., 2008; Chen et al., 2015; Aichner et al., 2019).

For the pollen transfer function (Fig. 10.A), the inferred reconstructions display similar trends during the 4500 years with larger amplitude for MAT than for WAPLS. MAT is also more sensitive to the initial calibration dataset selected than WAPLS (Fig. 10.A) which is in agreement with previous multi-methods studies (Brewer et al., 2008). Indeed, the COSTDB and EAPDB are over-reactive in front of local calibration, while the WAPLS display same amplitude for each calibration with a different offset. Particularly, the NMSDB consistently displays the values closest to the actual climate parameter values both for MAT and WAPLS. For the D3L6 precipitations, all models seem to drive away from the actual value. This shift highlights the *WorldClim2* interpolation issues in the Altai mountains (few weather stations and not accounting for snow melt).

On Fig. 10.B, about brGDGTs, the local calibrations (NMSDB applied to XRD section) provide the closest surface reconstructed temperature values to the actual ($\Delta$MAAT $< 2°$C, Fig. 10.B on the left panel) compared to the global calibration





**Figure 10.** ACA climate reconstruction for the 5000 year BP. (**A**): climate–pollen inferred for the Lake D3L6 (Unkelbach et al., 2019) comparing two transfer function methods (WAPLS and MAT) and the 4 databases (EAPDB, COSTDB, MDB and NMSDB) ; (**B**) climate–brGDGT inferred from the XRD section (Sun et al., 2019) comparing local (NMSDB) and global calibrations (De Jonge et al., 2014a; Naafs et al., 2017a, b). The climate periods correspond to Little Ice Age (LIA), Warm Medieval Period (WMP), Dark Ages Cold Period (DACP), Roman Warm Period (RWP) and 4.2k year event according to Zhang et al. (2008), Aichner et al. (2019) and Sun et al. (2019). Dashed lines represent the actual surface climate parameters foe each cores.





($\Delta$MAAT $\in [6;10]$°C, De Jonge et al., 2014a; Naafs et al., 2017a, b). This is explicable by the high similarity between the type of sediment from the XRD section (dry taphonomy) and a large proportion of the NMSDB surface samples (especially

from Gobi desert). Moreover, the NMSDB calibrations present a more realistic amplitude: 6 °C over 5000 years (similar to pollen-inferred amplitude) as opposed to 11 to 14 °C amplitude for global calibrations. About the precipitation, the brGDGTs mr–models show a decreasing trend along the Holocene, particularly well-marked between 1000 and 2000 yr BP followed by a bounce on the last 1000 years (tendency consistent with the WAPLS pollen MAP). All brGDGT calibrations exhibit consistent shifts during the LIA (cold-wet), the WMP (warm-dry) and the 4.2ka event (cold-wet). These variations are also exaggerated

with global calibrations. To conclude, general trends are consistent for all calibration datasets, except for the drying-warming trend inferred by the calibration from De Jonge et al. (2014a).

Figures 10.A and 10.B show that realistic reconstructed surface values are consistent with literature Holocene trends and validate the application of local calibrations for both pollen and brGDGTs. Furthermore, abrupt oscillations and overall ampli-

tudes of temperature and precipitation variations are realistic, in accordance with regional appraisal (Wu et al., 2020). These results permit us to improve our understanding on the Mongolia-China Late Holocene climate variations. Overall, on the 5000 year period the climate appear to follow a dryer-warmer trend. More precisely, both for pollen and brGDGTs the local calibration shows a correlation between temperature and precipitation short-period oscillations: the LIA seems to be colder and wetter than the warm and dry WMP. The same cold-wet behavior is observed for the 4.2ka event. This conclusion is important and

situate the D3L6 and the XRD sequences in the same trend that the majority of the ACA paleosequences (Chen et al., 2010, 2015; Wu et al., 2020), connecting the Altai range and the Qaidam basin to the EASM vs. Westerlies Holocene oscillations.

## 6   Conclusions

The palaeoenvironmental and palaeoclimatic signals may present several uncertainties (differential production, preservation...) which can misguide the interpretation of past variations. This study shows how both a multi-proxy approach and an accurate

calibration are important in preventing from these biases. We propose a new calibration for Mean Annual Precipitation (MAP) and Mean Annual Air Temperature (MAAT) from brGDGTs as well as a new pollen surface database available for transfer functions. The correlations between pollen rain and climate on one hand and brGDGT soil production and climate on the other are visible but are still mitigated by the complex climate system of Arid Central Asia and the diversity of soils and ecosystems. Precisely, each of our proxies seems to be more narrowly linked to precipitation (MAP) than temperature (MAAT) counter to

the majority of calibrations in the literature. This is validated on both modern and past sequences for pollen and brGDGTs. The nature of the samples considered (soil, moss polster and mud from temporary dry pond) also greatly affected these correlations. The calibration attempt for the extreme bio-climates of the Siberian basin and Mongolian plateau is difficult because of the low range of climate values, despite the climate diversity ranging from cold and slightly wet (north) to the arid and warm (south) conditions. Even if global and regional calibrations could be applied in such a setting, local calibrations provide enhanced

accuracy and specificity. The MAAT and MAP values do not remarkably spread in the vectorial space, which makes harder to



distinguish the linear correlation against variance noise. Moreover, this range of values is close to the lower saturation limit of the proxies, which makes the accurate local calibration tricky but necessary. The local calibrations also suffer from the reduced size and small geographic extent of the dataset. The vegetation cover, extending from a high cover taiga forest to bare soil desert cover, also buffers the climate signal and the GDGT / pollen response. The correlations between climate parameters and GDGT

/ pollen proportion are therefore lower than they could be at global scale. Nonetheless, and despite the lower correlation of the local calibration, these local approaches appear to be more accurate to fit the actual climate parameters than the global ones: both for pollen transfer functions and brGDGT multiple regression models. These positive model results have to be considered in light of over-parameterization limits. Too many parameters in mr–brGDGT models or in pollen MAT or WAPLS transfer function can add artificially to the linear relation between climate and proxies and lead to misinterpretation of palaeoclimate

records. Akaike's information criterion associated with RMSE and $R^2$ values is a fair way to select the best climate model. These local calibrations applied to D3L6 and XRD paleosequences highlighted the temperature and precipitation variation throughout the Late Holocene. The next step will be to test our calibrations on pollen and GDGTs records available from the same core. We encourage wider application of this local multi-proxy calibration for a more accurate constraint of these central Asian climatic systems, a crucial improvement to properly model the fluctuations of the Monsoon Line since the Holocene

Optimum.

*Author contributions.* LD conducted the analytical work, LD, SJ, OP, GM designed the study. All the authors contributed to the scientific reflection as well as to the preparation of the manuscript.

*Competing interests.* The authors declare having no competing interests.

*Acknowledgements.* We want to thank all the direct and indirect contributors to the global surface pollen dataset as well as the Laboratory of

Ecological and Evolutionary Synthesis of the National University of Mongolia for its support during the field trip. We also express gratitude to Laure Paradis for her GIS advice, Marc Dugerdil for the help with Python fixing, Jérôme Magail and the Monaco–Mongolia joint mission for their technical and financial support in providing top cores and sediment samples from Arkhangai, and Salomé Ansanay-Alex for her spectrometer expertise. We are grateful to the ISEM team DECG for financial support. For the analytical work completed at LGLTPE-ENS de Lyon, this research was funded by Institut Universitaire de France funds to GM.




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
