# Peer review of "Climate reconstructions based on GDGT and pollen surface datasets from Mongolia and Baikal area: calibrations and applicability to extremely cold-dry environments over the Late Holocene."

_Climate of the Past, 2020_

## Referee Comment (RC1) · Anonymous Referee #1 · 24 Jan 2021

The Fig. 9 upper part: the unit of of MAP is wrongly marked, and the why the MAP has negative data? It needs some technical checking.

---

## Referee Comment (RC2) · Anonymous Referee #2 · 1 Feb 2021

This study focuses on the calibration of proxy-climate relationships for pollen and GDGTs by comparing large published Eurasian calibrations with a set of 49 new surface samples. These calibrations are cross-validated by an independent dataset of top-core samples and then applied to two Late Holocene paleosequences from the Mongolian Altai and the Qaidam basin. GDGTs are relatively new proxies for reconstructing environmental conditions and any studies on calibrating modern datasets relative to climate data is very important. Along those lines, this work is interesting from a methodological point of view. However, the choice of paleodata for the application of the

results obtained in the calibration is not correct: none of the paleo records does belong to the territory of the Mongolian Plateau described in the paper. Another disadvantage is that the data on pollen and on GDGTs were obtained from different paleosequences. Of course, not much data is currently available on GDGTs from Mongolia. Therefore, we will not consider this a shortcoming of the study, but we will consider it a consequence of the little research of the territory. However, pollen data from Mongolia are abundant. The next point is the definition of the study area. The map shows the entirety of Mongolia, with the Mongolian Plateau not highlighted. Part of the locations is near Lake Baikal (the authors call it the Siberian Basin). First of all, there is no Siberian Basin and there are mountains around Baikal. According to the map, samples were taken only along the Angara River, where the height is about 700 m asl. This location is only a minor point in the huge territory of Siberia. I propose to work out the geography of the study. Try to prove that the results of the calibration of your samples taken from the Angara and then the transect to the south can really be applied to the sites located in the Mongolian Altai and Qaidam. The ideal would have been to choose paleosites within your transect. At least try to explain why this is not being done. I also suggest adding South Siberia or Baikal area in the title instead of Siberia.

Specific comments Lines 14-15: "(3) even if local calibrations suffer from reduced amplitude of climatic parameter due to local homogeneity, they better reflect 15 actual climate than the global ones by reducing the limits for saturation impact," This statement is pretty obvious without many pages of statistics. Lines 65-66: "Environmental drivers are linked to climate parameters (Weijers et al., 2007b), soil typology and vegetation cover (Davtian et al., 2016), which in turn imply land cover and land use" I don't understand to what link this statement. To BrGDGT? Line 75: "CBT and MBT indexes". What is this? Abbreviations appear for the first time, they must be deciphered. Lines 76-78: "the 5-methyl correlates mainly with temperature (Naafs et al., 2017a), while 6 and 7-methyl seem to react to moisture and pH" What do you mean, "correlate," "react"? Is there something happening to them that is quantified? Lines 80-84: "MBT05Me, Index1, Ri/b" Are you sure that all readers of your paper know what is this? I don't know,

for example. Line 115: "Fig. 1.A)." I don't see Siberia on this map. Lines 120-121: "from the Siberian Oblast of Irkoustsk, Russia" No such administrative or geographical district in Russia. Line 129: cyperaceae should be Cyperaceae Lines 136-138: Strange choice of locations: both do not belong to the Mongolian plateau Line 147: "the geography is characterized by the Baikal lake basin" What do you mean? Line 153: The dark-taiga cannot be dominated by larches (Larix sibirica). Larch is dominant of light taiga. Line 164: Here should be references concerning vegetation description. Line 165: "2.3 Bioclimate Systems" Do you mean 'climate'? Lines 174-175: "Mongolian summer precipitations are controlled by the East Asian Summer Monsoon system (EASM) instead of the Westerlies'. Please provide references that in Northern Mongolia and the Altai Mountains are the summer precipitation controlled by EASM. I think it is not correct and even your Fig. 2 supports this. Line 206: "3.2 SIG Bioclimatic Data". What is SIG? Line 245: "Statistical Analyses", but in lines 227-244 was also statistical analysis. Line 260, 279: "Siberian basin" is absent on the geographical map. Lines 263-264: Fig 5 appears before Fig 4. Lines 313-314: This information about IsoGDGTs you can move to the part when you describe the brGDGTs in the introduction Line 350: What is NSSM? This paper is going to publish in a journal with a wide readership. You have to explain specific terms. Line 367: Fig. 8, appears before Fig. 7 Line 403: "the Siberian-Mongolian system seems to be mainly controlled by precipitation". I just want to emphasize that Siberia has a huge size and varied landscapes. Lines 480: There is not Baikal basin on the geographical map. Baikal is divided into three basins, but it is about the lake. Lines 498, Fig. 9: NSMDB – wrong name Fig. 9: Negative values of MAP should be explained. Fig. 10. The chronological curve is filled only up to 5000 yr BP. Remove the blank part up to 6000 yr BP.

---

## Referee Comment (RC3) · Anonymous Referee #3 · 25 Feb 2021

This paper presented valuable data sets of pollen assemblage and bacterial branched GDGTs from Mongolia and Siberia that are both extremely dry and cold regions. Regional or local calibrations for temperature and precipitation reconstruction were developed based on the relationship of pollen assemblages and the distribution of brGDGTs with climatic parameters. The authors then used two sediment profiles published in other studies to validate their proposed calibrations and posited that the calibrations were applicable to the cold and dry regions, as stated in the title. This study is valuable in terms of the scarcity of pollen assemblage and bacterial brGDGT data from Mongolian and Siberian soils. This paper is generally well written. But the phrases or wording in this paper need to be improved, as can be seen below. I also have some concerns that need to be clearly addressed. First, the authors have mentioned 7-methyl brGDGTs that were not widely present in soils. So I am not so convinced that the compounds they identify in these soils are 7-methyl brGDGTs since they do not provide any useful information to support this. This, however, needs to be accurate. Second, the paper discussed two types of environmental indicators: pollen and brGDGTs, thus providing abundant information. However, in other words, the paper is rather complex and unfocused. I would write pollen and brGDGTs in two separate papers. The combination of them into one paper results in insufficient discussions. For example, I do not see many discussions about the pollen and mechanisms behind the environmental control on brGDGTs and pollen in such extremely cold and dry regions. The response of pollen and brGDGTs to precipitation and temperature was dependent on biochemical mechanisms. The model or calibrations developed by statistical methods should be consistent with these mechanisms. Third, the authors developed calibrations based on brGDGTs in Mongolian soils; however, no paleo sequences or loess-like sediments can be used in this region to assess the applicability of these calibrations. So I questioned the value of these calibrations. Fourthly, different types of sediments were used in this paper, e.g., muds, soils, and moss. I learn from previous studies that brGDGTs might have different sources in different sediments. So we can see calibrations specific for each environment have been proposed over the last decade. I am very confused about the use of all these sediments in the development of a calibration. Other concerns are listed below for your consideration.

Line 34 '?' Question mark. References needed here. Line 46 Change 'leads' to 'lead' Line 57 Redundant. Pls delete 'and agree well'. Line 63 'Damsté et al., 2000' should be 'Sinninghe Damsté et al., 2000' Line 69 'Salvador-Castel et al., 2019 in press' Pls list this reference in the reference list. Line 71-72 'bacterial community structure (Xie et al., 2015), the bacterial group response(Knappy et al., 2011) and the GDGT occurrences in different bacterial communities (Liu et al., 2012b) to. All these references

are related to archaea that completely differ from bacteria. They are archaeal community and archaeal group, not bacterial community. Line 75 MBT and CBT need to be defined since they appear for the first time. Line 76-77, 78 'reacts' better use 'respond'. Add brGDGTs after '5-methyl', add '-' after '5,6' Line 78 Add brGDGTs after '6-methyl'. Line 84 'Ri/b' should be defined upon its first occurrence in the text. Line 130-135 Mud from ponds generally contains GDGT distribution that differs markedly from neighboring soils. The calibration for pond-like sediments is also different from that of soils. I cannot agree with the incorporation of mud sediments in developing calibrations for soil environments. Line 171 'precipitations' changed to 'precipitation'. Line 224 APCI needs to be explained for the wide readership of the journal. What is 'LGLTPE-ENS de Lyon'? Line 226-230 I found from the Result part (figure 4) that the authors have identified a series of 7-methyl brGDGTs, which were not widely seen in soils and lakes. Please provide the details as to how these compounds were identified and assigned in the text so that reviewers can assess whether they are identified in a right way. Line 261 and Figure 3 captions What are 'AP' and 'NAP'? Please define this term prior to use. Line 279-280 Please show the determination coefficients and p values for the correlations Table 1 It is a little bit confusing to see so many abbreviations in the table. Please provide notes below the table. Line 312 '74.6%' I think this may be the average abundance of each compound for all the soil samples. Please specify. Line 316 diverge from surface soil samples? Please make it clear. Line 316-317 The same iGDGT distribution between soil and lake sediments does not necessarily indicates a significant contribution of GDGTs to the lake. GDGT-0 dominates over crenarchaeol in these soils, probably reflecting a high alkalinity of the soils or a dominance of methanogenic Euryarchaeota due to the anoxic environments in the moss. In contrast, the dominance of GDGT-0 over crenarchaeol in lake sediments might indicate that abundant methanogenic Euryarchaeota inhabit the anoxic lake sediments yet few Thaumarchaeota live in the lake. Line 317 'soil-produced' changed to 'produced in soils' Line 318 '[crenarcheol]' What does the bracket indicate? Please specify in the text. Line 319 What does 'reaction' mean? Line 327 Reduce '22.77%' to one significant

figure. Line 341 'methyled and cyclized' changed to 'methylated and cyclized' Line 349 Add '-' after '5 and 6'. Line 376 Add 'brGDGTs' after '6-methyl'. Line 396 O2 Line 417 'react' Not properly used word. Line 442 'affects' changed to 'affect'. Line 502 extreme Figure 10 captions 'foe each cores' 'for each core'. Line 567 drier

---

## Author Comment (AC1) · 3 Mar 2021

**1   Responses to the comments of Reviewer 1 (Anonymous Referee)**

**1.1   Specific comments:**

The Fig. 9 upper part: the unit of of MAP is wrongly marked, and the why the MAP has negative data? It needs some technical checking.

[Figure]

**Response:** To check the technical problems we found 12 sites with negative precipitation values in our modern pollen dataset. It appears that these points are located in Arid Central Asia (ACA), a region where the weather stations are very scarce. This could explain the problem of interpolation for the climatic parameter selection. A new interpolation applied to these points allows to obtain now more reliable precipitation values.

**Applied changes:** The legend (Fig.9, or Rebuttal Fig. 1) has been changed to $MAP(mm.yr^{-1})$. For the figure 9 or Rebuttal Fig. 1, we have plotted the figure with new interpolation values for the 12 previously dysfunctional points.

**Fig. 1.** Relationships between the eight major pollen taxa (\%TP) and MAP (mm.yr^-1, upper part of the facet plot) and MAAT (°C, lower part).

---

## Author Comment (AC2) · 3 Mar 2021

**1   Responses to the comments of Reviewer 2 (Anonymous Referee)**

**1.1   General comments:**

This study focuses on the calibration of proxy-climate relationships for pollen and GDGTs by comparing large published Eurasian calibrations with a set of 49 new

surface samples. These calibrations are cross-validated by an independent dataset of top-core samples and then applied to two Late Holocene paleosequences from the Mongolian Altai and the Qaidam basin. GDGTs are relatively new proxies for reconstructing environmental conditions and any studies on calibrating modern datasets relative to climate data is very important. Along those lines, this work is interesting from a methodological point of view.

However, the choice of paleodata for the application of the results obtained in the calibration is not correct: none of the paleo records does belong to the territory of the Mongolian Plateau described in the paper.

**Response:** The paleo-validation of a calibration study is always difficult in an area where there are still a few paleo studies. This is the case for the Mongolian plateau. More over, the definition of the Mongolian Plateau (MP) borders is changing from a study to another (Windley et al., 1993; Meng et al., 1998; Wang et al., 2013; Sha et al., 2015; Chen et al., 2015a). However for the majority among them, the MMNT5C03 top-core and the D3L6 core are within the Mongolian Plateau. For the GDGT sequence, the only available lake dataset was collected on the Loess Plateau which is the closest geographical plateau to the MP. Therefore, the environmental conditions on the Loess Plateau (elevation, climate parameters, vegetation...) are similar to the ones prevailing on the MP.

**Applied changes:** We do agree that the paleo-validation is more accurate when the paleo-sequence is close to the surface sample transect. That is why we have applied our calibration on an other pollen sequence from Dulikha Bog (Bezrukova et al., 2005; Binney 2017) which is very close to the Lake Baikal in the Buriakya Republic (Rebuttal Figs. 2 and 3).

Another disadvantage is that the data on pollen and on GDGTs were obtained from different paleosequences. Of course, not much data is currently available on GDGTs from Mongolia. Therefore, we will not consider this a shortcoming of the study, but we will consider it a consequence of the little research of the territory. However, pollen data from Mongolia are abundant.

**Response:** We fully agree with this comment. The multi-proxy validation approach should be conducted on the same core. But there is, to our knowledge, no multi-proxy study available to date in Arid Central Asian (ACA) area. Such a comparative multi-proxy calibration study have to be conducted in the next years.

**Applied changes:** For the time being, we found a recently brGDGT paleosequence (NRX) with open access data from (Rao et al., 2020). This sequence is perfect for our calibration validation test : the peat core come from Altai mountains, with an elevation (around 1700 m a.s.l) close to the NMSDB average elevation and with same range of climate parameters, is only 200 km away from the D3L6 sequence as the crow flies. Therefore in the revised manuscript, we were able to compare the brGDGT calibration on two paleo-sequences (NRX and XRD) and to compare the brGDGT and the pollen signal for Altai mountains (NRX and D3L6). The location of the new paleo-sequences have been added to Rebuttal Figs. 2 and 3 and the results are displayed in Rebuttal Fig. 1.

The next point is the definition of the study area. The map shows the entirety of Mongolia, with the Mongolian Plateau not highlighted.

**Applied changes:** We have changed the figure 2 or Rebuttal Fig. 2 to highlight the MP following the geological definition (Windley et al., 1993; Sha et al., 2015) and the

political borders (Meng et al., 1998, Chen et al., 2015a) including the whole Inner and Outter Mongolia plus northern part of the Xinjiang and Gansu provinces and the Buriakya and Tuva Republic, the Zabaykalsky Krai and the Amur Oblast.

Part of the locations is near Lake Baikal (the authors call it the Siberian Basin). First of all, there is no Siberian Basin and there are mountains around Baikal. According to the map, samples were taken only along the Angara River, where the height is about 700 m asl. This location is only a minor point in the huge territory of Siberia. I propose to work out the geography of the study.

**Response:** We definitely acknowledge that the word Siberia is not precise enough to describe the possible range of application of our calibration dataset. Indeed, our surface samples were collected from the wide definition of the Mongolian Plateau, that is to say the Baikal area, the Khentii and Khangai mountains, the central Mongolian steppes and the Gobi desert.

**Applied changes:** We have changed Siberia by Baikal Area in the text. Also, the location of Baikal and Angara have been added on the map Rebuttal Fig. 3.

Try to prove that the results of the calibration of your samples taken from the Angara and then the transect to the south can really be applied to the sites located in the Mongolian Altai and Qaidam. The ideal would have been to choose paleo-sites within your transect. At least try to explain why this is not being done.

**Applied changes:** Two new past records within the transect have been added and discussed into the 5.2.4 paragraph. The reconstructed climate parameter trend follows the previously used paleo-sequence reconstruction (from D3L6 and XRD) trend.
I also suggest adding South Siberia or Baikal area in the title instead of Siberia.

**Applied changes:** The title has been changed into *Climate reconstructions based on GDGT and pollen surface datasets from Mongolia and Baikal area: calibrations and applicability to extremely cold-dry environments over the Late Holocene.*

1.2   Specific comments:

Lines 14-15: "(3) even if local calibrations suffer from reduced amplitude of climatic parameter due to local homogeneity, they better reflect actual climate than the global ones by reducing the limits for saturation impact," This statement is pretty obvious without many pages of statistics.

**Response:** The saturation impact is a tricky question currently discussed in brGDGT studies associated to extreme climate reconstruction, that is why it appears mandatory to insist on it.

Lines 65-66: "Environmental drivers are linked to climate parameters (Weijers et al., 2007b), soil typology and vegetation cover (Davtian et al., 2016), which in turn imply land cover and land use" I don't understand to what link this statement. To BrGDGT?

**Response:** This sentence summarizes the major environmental factors than can drive the brGDGT assemblages.

**Applied changes:** The sentence (L. 68) has been changed into BrGDGT environmental drivers are linked to (...) land cover and land use.

Line 75: "CBT and MBT indexes". What is this? Abbreviations appear for the first time, they must be deciphered.

**Applied changes:** The line 78 has been changed into To monitor these changes, Cyclisation ratio of Branched Tetraethers (CBT) and Methylation index of Branched Tetraethers (MBT) indexes linked to environmental factors such as climate and soil parameters have been proposed (Weijers et al., 2007b; Huguet et al., 2013a)

Lines 76-78: "the 5-methyl correlates mainly with temperature (Naafs et al., 2017a), while 6 and 7-methyl seem to react to moisture and pH" What do you mean, "correlate," "react"? Is there something happening to them that is quantified?

**Response:** These two propositions have indeed been mathematically shown but the actual ecophysiological process behind these correlations are still not well understood.

**Applied changes:** The sentence has been modified to underline the mathematical aspect of the relations and the R-squared values of the studies have been added. We have (L. 80-84) the 5-methyl mathematically correlates mainly with temperature ($R^2 = 0.76$, Naafs et al., 2017), while 6 ($R^2 = 0.69$) and 7-methyl ($R^2 = 0.44$) seem to moderately correlates with moisture and pH (Yang et al., 2015; Ding et al., 2016).

Lines 80-84: "MBT05Me, Index1, Ri/b" Are you sure that all readers of your paper know what is this? I don't know, for example.

**Response:** We do agree that this indexes are very technical and precise. But, within the brGDGT research communities the question of the accuracy of these indexes is determinant. To make this part more fluid and digestible, we try to deciphered each acronym.

**Applied changes:** The sentence L. 83-86 has been changed in More specific indexes have been proposed by DeJonge et al., (2014b) to limit the multi-correlation systems with the withdrawal of 5-methyl compounds such as $MBT'_{5Me}$ which is independent of the pH and $CBT_{5Me}$ which is more representative of the soil pH than the former version of the index (index formula in Supplementary Table S1). . The sentence L. 91 has been changed in The Ratio of isoGDGT on brGDGT ($R_{i/b}$) has been proposed as a reliable aridity proxy (Yang et al., 2014; Xie et al., 2012) .

Line 115: "Fig. 1.A)." I don't see Siberia on this map.

**Applied changes:** Siberia  has been changed to Baikal area  on lines 33, 105, 123, 130, 269, 287, 307, 412, 415, 416, 505, 530 and added on the map.

Lines 120-121: "from the Siberian Oblast of Irkoustsk, Russia" No such administrative or geographical district in Russia.

**Response:** We thought the eastern edge of the Lake Baikal was under the Irkutsk Oblast district.

**Applied changes:** The sentence (L. 130) has been modified into a fifth transect has been done in the Sayan range along the Angara valley, Russia (*MRUT1, n = 12,* on

Fig. 1.G).

Line 129: cyperaceae should be Cyperaceae

**Applied changes:** Modified accordingly.

Lines 136-138: Strange choice of locations: both do not belong to the Mongolian plateau

**Applied changes:** The sentence (L. 149) has been modified into For the pollen analysis, the cores D3L6 from Unkelbach et al. (2019) located in the Mongolian Altai range and the Dulikha bog located in the Sayan range, Baikal area (Fig. 1, Bezrukova et al., 2005; Binney, 2017) are compared to the Xiangride section (XRD) used for brGDGT sequence from Sun et al. (2019), sampled in the Chinese Qaidam Basin and the NRX peat bog (Chinese Altai, Fig. 2, Rao et al., 2020).

Line 147: "the geography is characterized by the Baikal lake basin" What do you mean?

**Applied changes:** The sentence (L. 160-162) has been changed into In the northern-most part of the MP, the Baikal lake area is characterized by a basin at a lower altitude (around 600 m a.s.l, Fig. 3.G, Demske et al., 2005).

Line 153: The dark-taiga cannot be dominated by larches (Larix sibirica). Larch is dominant of light taiga.

**Response:** Indeed, we tried to apply these sub-taiga vegetation community to MP

vegetation communities which is not right. We corrected this confusion using the description of both Demske et al., (2005) and Schlutz et al., (2008). Finally, they describe a Light taiga-riparian forest community in the Angara valley and a Mixed light/dark taiga–birches sub-taiga on the MP.

**Applied changes:** Finally, these vegetation communities have been described as following (L.164-166) The distribution of vegetation and biomes follows a latitudinal belt organization: in the North, the boreal forest presents a mosaic of light-taiga dominated by *Pinus sylvestris* mixed with riparian forest dominated by birches (*Betula* spp.), alders (*Alnus* spp.) and willows (*Salix* spp., Demske et al., 2005). On the MP, the light-taiga dominated by larches (*Larix sibirica*) and few birches is mixed with dark-taiga composed of Siberian pines (*Pinus sibirica*) and spruces (*Picea obovata*, Schlütz et al., 2008). Moreover, the Figs. 3 and 5 as well as the result paragraph 4.1.1 have been actualized following these new community definitions.

Line 164: Here should be references concerning vegetation description.

**Applied changes:** We used Demske et al., (2005), Dulamsuren et al., (2005b), Schlutz et al., (2008) and Klinge et al., (2018) as reference for the actual vegetation.

Line 165: "2.3 Bioclimate Systems" Do you mean 'climate'?

**Response:** The use of the *Bioclimate* term allows to discuss the influence of climate systems on the vegetation communities of the MP.

Lines 174-175: "Mongolian summer precipitations are controlled by the East Asian

Summer Monsoon system (EASM) instead of the Westerlies'. Please provide references that in Northern Mongolia and the Altai Mountains are the summer precipitation controlled by EASM. I think it is not correct and even your Fig. 2 supports this.

**Response:** The question of the climate system influencing the MP is a tricky point still under debate (Piao et al., 2018). It seems that the westerlies dominate the surrounding areas but it is still paradoxical that almost all the precipitation occurs in summertime almost on the whole MP (Dulamsuren et al., 2005b) which is more associated with the EASM system. The northern part of this EASM front explain the precipitation amount up to the Baikal area (Shukurov et al., 2017 and periodic phenomenon (weak EASM involves anticyclone on the MP) could also explain the northwest set up of this front onto ACA and MP (Zhang et al., 2021). In any case, Piao et al., (2018) insist on the impact of the local water evaporation on the recycling moisture in MP and Siberia.

**Applied changes:** The sentence L. 191-193, has been changed and enriched as follows An unknown amount of precipitation occurs in winter as snowfall (Rudaya et al., 2020) which it is not always measured into the weather station MAP. The main part of the MP MAP occurs during the summer (climate diagrams from Dulamsuren et al., 2005). However, the precipitation origin for Mongolia is still under debate (Piao et al., 2018). Mongolian summer precipitations up to the Baikal area (Shukurov and Mokhov, 2017) seem to be controlled by the East Asian Summer Monsoon system (EASM) instead of the Westerlies' winter precipitation stocked onto the Sayan and Altai range (Fig. 2; An et al., 2008). The alternating Westerlies / EASM domination on the MP climate system appears to fluctuate throughout the Holocene depending on the monsoon strength (Zhang, 2021): the weakest is the monsoon, the furthest the EASM bring precipitation up to the ACA hyper-continental area. The EASM force may variate in function of the MP snow cover (albedo effect on sun radiance impact, Liu and Yanai, 2002) and/or the Pacific surface temperature (Yang and Lau, 1998). Finally,

Piao et al. (2018) insist on the locally evaporated water recycling importance within the Mongolian MAP amount.

Line 206: "3.2 SIG Bioclimatic Data". What is SIG?

**Response and applied changes:** SIG is the french version of a GIS. We have corrected this translation mistake.

Line 245: "Statistical Analyses", but in lines 227-244 was also statistical analysis.

**Response and applied changes:** The part 3.3 is about GDGT analysis and the way we can connect brGDGT abundances with climate parameters. The part 3.4 is more about the general tools used in the study to describe the variance of the different data sets. To make it clearer the part 3.3 has been changed into GDGT Analysis and Calibrations .

Line 260, 279: "Siberian basin" is absent on the geographical map.

**Response and applied changes:** Siberian basin  has been changed into Baikal area (L. 33, 105, 123, 130, 269, 287, 307, 412, 415, 416, 505, 530).

Lines 263-264: Fig 5 appears before Fig 4.

**Response:** The Fig. 4 is called line 245 for the fist time, the Fig. 5 appears after it, in line 280.

Lines 313-314: This information about isoGDGTs you can move to the part when you describe the brGDGTs in the introduction

**Response and applied changes:** The sentence has been moved to the introduction (L. 87).

Line 350: What is NSSM? This paper is going to publish in a journal with a wide readership. You have to explain specific terms.

**Response and applied changes:** The $N_{SSM}$ value is the number of different Stepwise Selection Model (SSM) possible to calculate. We have reminded on the previous sentence (L. 375) the meaning of the SSM acronym.

Line 367: Fig. 8, appears before Fig. 7

**Response and applied changes:** The two figures have been interchanged to follow the text development.

Line 403: "the Siberian-Mongolian system seems to be mainly controlled by precipitation". I just want to emphasize that Siberia has a huge size and varied landscapes.

**Response and applied changes:** Here too, we changed the sentence in Baikal area-Mongolian Plateau . In the paragraph, we add also Southern to the Siberian-Mongolian system. The labels in the former Fig. 8 (newly Fig. 7) have also been modified from Siberian to Baikal .

Lines 480: There is not Baikal basin on the geographical map. Baikal is divided into three basins, but it is about the lake.

**Response and applied changes:** Each Baikal basin have been replaced by Baikal area .

Lines 498, Fig. 9: NSMDB – wrong name

**Response and applied changes:** Modified accordingly, as well as in the Fig. 9 label.

Fig. 9: Negative values of MAP should be explained.

**Response and applied changes:** This point was addressed in the Referee 1 comments responses and we applied changes in the text.

Fig. 10. The chronological curve is filled only up to 5000 yr BP. Remove the blank part up to 6000 yr BP.

**Response and applied changes:** Modified accordingly.

* * *
[Figure]

**Fig. 1.** ACA climate reconstruction for the 5000 year cal BP. A: climate–pollen inferred for the Dulikha and Lake D3L6. B: climate–brGDGT inferred MAAT and MAP.

[Figure]

**Fig. 2.** Eurasian map of all the pollen surface samples included in the database. The color code refers to the biome pollen inferred for each site.

**Fig. 3.** A: Topographic map of Mongolia (from ASTER data) with the location of surface samples and weather stations considered in the present study; B: MAP; C: MAAT; D: Gobi desert focus; E: Khentii focus

---

## Author Comment (AC3) · 3 Mar 2021

**1   Responses to the comments of Reviewer 3 (Anonymous Referee)**

**1.1   General comments:**

This paper presented valuable data sets of pollen assemblage and bacterial branched GDGTs from Mongolia and Siberia that are both extremely dry and cold regions.

Regional or local calibrations for temperature and precipitation reconstruction were developed based on the relationship of pollen assemblages and the distribution of brGDGTs with climatic parameters. The authors then used two sediment profiles published in other studies to validate their proposed calibrations and posited that the calibrations were applicable to the cold and dry regions, as stated in the title. This study is valuable in terms of the scarcity of pollen assemblage and bacterial brGDGT data from Mongolian and Siberian soils. This paper is generally well written. But the phrases or wording in this paper need to be improved, as can be seen below.

I also have some concerns that need to be clearly addressed. First, the authors have mentioned 7-methylbrGDGTs that were not widely present in soils. So I am not so convinced that the compounds they identify in these soils are 7-methyl brGDGTs since they do not provide any useful information to support this. This, however, needs to be accurate.

**Response:** We do agree that the method has to be accurately described and especially when it concerns quite recent proxy improvement such as 7-methyl brGDGT detection. To identify the 7-methyl brGDGT compound we have followed the approach described by Ding et al., (2016) for the peak identification for the Cameroon lake chromatograms. It is principally based on the m/z and relative retention time of each compound. About the slight proportion of 7-methyl brGDGTs in the Mongolian soil, it appears that we obtained an average of 4.6 % in Mongolian soil against 6.2% for Chinese lakes and 4.3% in Cameroon lakes (Ding et al., 2016). We observe same order of magnitudes.

**Applied changes:** The sentence L.255 in the method has been modified as follow : " Each compound was identified and manually integrated according to its m/z and relative retention time following the integration descriptions from Liu et al., (2012b)

and DeJonge et al., (2014b) for 5- and 6-methyl brGDGT and Ding et al., (2016) for 7-methyl brGDGTs." Then, in the result part we have appended this observation (L. 366) " Even if the 7-methyl brGDGTs appear to have weak significance in the brGDGT variance explanation (Fig. 5.A), the surface samples 7-methyl average fractional abundance around 4.6% is following the normal order of magnitude (4.3% in Cameroon lakes and 6.2% for Chinese lakes, Ding et al., 2016). "

Second,the paper discussed two types of environmental indicators: pollen and brGDGTs, thus providing abundant information. However, in other words, the paper is rather complex and unfocused. I would write pollen and brGDGTs in two separate papers. The combination of them into one paper results in insufficient discussions. For example, I do not see many discussions about the pollen and mechanisms behind the environmental control on brGDGTs and pollen in such extremely cold and dry regions. The response of pollen and brGDGTs to precipitation and temperature was dependent on biochemical mechanisms. The model or calibrations developed by statistical methods should be consistent with these mechanisms.

**Response:** We can understand that this article appears to be complex because it is going into technical issues of two different proxy methodologies. The strength of this study is precisely to promote an interdisciplinary approach. It is addressed to two different communities: the organist geochemist and the palynologist. These two scientific communities are working to the same goal (the past environment and climate understanding) with tools slightly different but based on same hypothesis and methods. The topic of this article is precisely comparing the two calibration processes on the same dataset. It is the first time that this pollen-GDGTs interdisciplinary approach is proposed and tested on the same calibration. This article addresses a review of biases of both proxy and solutions to take them into account. This approach aims to help geochemists and palynologists researchers to have a clearer idea of the

benefits and flaws of each proxy.

By the way, we thank the three referees (coming from the two different communities) for all of their accurate comments which improve the readability of this article: language corrections, better method description, archaeal communities description, acronym definition, significant figure reducing, etc. For the discussion on mechanisms controlling the pollen and brGDGT distribution, we agree that this question is a critical issue, but it is not exactly the topic of this article since we are presenting mathematical calibrations and not pollen estimation production (such as PPE approach, Ge et al., 2017; Li et al., 2017) or brGDGT producing community (since we are not working on the archaeal soil communities in this article). This topic will be the object of a subsequent detailed article (Dugerdil et al., in prep.). As described in the introduction, this article applies a black box mathematical models without consideration to ecophysical mechanism in a first step. Then, the models are validated by such mechanisms.

**Applied changes:** To refer to the importance of such works the sentence " In any cases, a better understanding of the archaeal community responses to ecophysiological parameters variations will considerably improve the brGDGT calibration process (Xie et al., 2015; Dang et al., 2016; De Jonge et al., 2019). " has been added (L. 440), we also precised that " the effects of frozen soils on soil archaeal communities and GDGT production are thought to be important (Kusch et al., 2019) since the archaeal community seems to be shifting with abrupt temperature modifications (De Jonge et al., 2019)." (L. 453).

Third, the authors developed calibrations based on brGDGTs in Mongolian soils; however, no paleo sequences or loess-like sediments can be used in this region to assess the applicability of these calibrations. So I questioned the value of these calibrations.

**Response:** This comment has been formulate by the referee 2 too. We applied the calibration to two more sequences one in Baikal area (Dulikha, Bezrukova et al., 2005; Binney, 2017) and one in the Chinese Altai (NRX, Rao et al., 2020). Please report to the answer addressed to referee 2. Moreover, this calibration study is a preliminary work for the brGDGT-inferred climate reconstruction made on Mongolian lakes, a region poorly documented by molecular biomarkers records. Then, the MP could be an efficient analogue for other cold semi-arid areas in central Asia (in Altai, Tian Shan, Qilian Shan, Sayan range...) because of the similitude in vegetation (Erds et al., 2018, Klinge et al., 2018) and in climate history (Zhang 2021). The NRX sequence (added following the advice of Referee 2) from the Altai mountains is a good example (Rao et al., 2020).

**Applied changes:** For the applied change, please report to the referee 2 answer. To sum-up, we have added a new paleo-sequence from the Chinese Altai range which is closer from the surface data set presented in this study into the discussion. Also the rebuttal figures 1, 2 and 3 have been modified with this new paleo-sequence.

**Link to Referee 2 response and applied changes:** For the time being, we found a recently brGDGT paleosequence (NRX) with open access data from (Rao et al., 2020). This sequence is perfect for our calibration validation test : the peat core come from Altai mountains, with an elevation (around 1700 m a.s.l) close to the NMSDB average elevation and with same range of climate parameters, is only 200 km away from the D3L6 sequence as the crow flies. Therefore in the revised manuscript, we were able to compare the brGDGT calibration on two paleo-sequences (NRX and XRD) and to compare the brGDGT and the pollen signal for Altai mountains (NRX and D3L6). The location of the new paleo-sequences have been added to Rebuttal Figs. 2 and 3 and the results are displayed in Rebuttal Fig. 1.

Fourthly, different types of sediments were used in this paper, e.g., muds, soils, and moss. I learn from previous studies that brGDGTs might have different sources in different sediments. So we can see calibrations specific for each environment have been proposed over the last decade. I am very confused about the use of all these sediments in the development of a calibration. Other concerns are listed below for your consideration.

**Response:** We do agree that the type of environment determines the composition in brGDGT fractional abundances. This observation is particularly true for deep lake / marine environment / surface soil samples differentiation. However, if the in-situ production of brGDGT is strongly enhanced in wide and deep lakes, it seems not to be the case for the two temporary pond mud samples from the surface data-set. Following Pearson et al., (2011a) only the samples with BIT index < 0.4 are really following a lake brGDGT signals. On the Rebuttal Fig. 4, we can see that our 2 pond muds samples really follow a soil produced brGDGT signal. About the IIIa/IIa ratio, Cao et al., (2020) find values below 0.86 for soil and values above 1.15 for northern arid Chinese lake sediment. This ratio (Xiao et al., 2016; Martin et al., 2019a) is used to indicate the brGDGT origin. Here too (Rebuttal. Fig. 4) the IIIa/IIa ratio indicates that these two samples are controlled by brGDGT-soil production. The other sediment samples (MMNT5C11 and MMNT5C12) are not used into the calibration dataset, they are just displayed to make comparison with the surface database. In Mongolia, we do think that the surface sample type heterogeneity (soil, moss, pond mud) is not impacting the calibration reliability because the area is very arid. Even a small climate/environment change in precipitation amount can convert a lake into pond or into wet meadow including moss polster. Thus, the archive type is also varying in between these types of environment. Calibrations specific for each environment are most interesting at the global scale but not at the local scale where the surface

dataset should be representative of the local variability. Finally, we do think that the bias induced by this samples heterogeneity is smaller than the bias induced by tiny dataset.

**Applied changes:** To made this more clear the label of Fig. 4 and Fig. 6 have been changed from " Lacustrine " to " Pond mud ". The sentence (L.281) has been precised with " A cross-validation test was performed for all the brGDGT calibrations (from this study and from the literature) using an independent set of six lacustrine samples from the lake *MMNT5C12* top-core. " We also have added the Rebuttal Fig. 4 in Supplementary information S1. And the following sentences have been introduced into the manuscript " The GDGT input origin could be traced using the BIT index (Branched and Isoprenoid Tetraether index, Hopmans et al., 2004; Pearson et al., 2011a) and the $III_a/II_a$ ratio (Xiao et al., 2016, Martin et al., 2019a, Cao et al., 2020). " (L. 68) and " About the pond mud samples, the BIT and $III_a/II_a$ ratio (Supplementary Fig. 4) show that a coherent soil origin is leading the brGDGT input instead of a lacustrine one (Pearsonet al., 2011; Martin et al., 2019a; Cao et al., 2020). " (L. 376). To clarify the SSM method we also added " The SSM has been applied on the total surface dataset (including the pond muds validated by the Supplementary Fig. S1 " (L. 394)

**1.2 Specific comments:**

Line 34 '?' Question mark. References needed here.

**Response and applied changes:** The Braconnot et al., 2021a reference was miss-indexed into the bibliography index. The error has been corrected.

Line 46 Change 'leads' to 'lead'
Line 57 Redundant. Pls delete 'and agree well'.

Line 63 'Damsté et al., 2000' should be 'Sinninghe Damsté et al., 2000'

**Response and applied changes:** Texts and reference modified accordingly.

Line 69 'Salvador-Castel et al., 2019 in press' Pls list this reference in the reference list.

**Response and applied changes:** The published version of the article has been actualized in the reference list.

Line 71-72 'bacterial community structure (Xie et al., 2015), the bacterial group response(Knappy et al., 2011) and the GDGT occurrences in different bacterial communities (Liu et al., 2012b) to. All these references are related to archaea that completely differ from bacteria. They are archaeal community and archaeal group, not bacterial community.

**Response and applied changes:** Text modified accordingly. Each occurrence of " bacterial " in the text have been change by " archaeal " (L. 67, 75, 76, 107, 438 and 454).

Line 75 MBT and CBT need to be defined since they appear for the first time.

**Response and applied changes:** The text has been modified following the comment of Referee 2.

Line 76-77, 78 'reacts' better use 'respond'. Add brGDGTs after '5-methyl', add '-' after '5,6'

**Response and applied changes:** Text modified accordingly.

Line 78 Add brGDGTs after'6-methyl'.

**Response and applied changes:** Text modified accordingly.

Line 84 'Ri/b' should be defined upon its first occurrence in the text.

**Response and applied changes:** The text has been modified following the comment of Referee 2.

Line130-135 Mud from ponds generally contains GDGT distribution that differs markedly from neighboring soils. The calibration for pond-like sediments is also different from that of soils. I cannot agree with the incorporation of mud sediments in developing calibrations for soil environments.

**Response and applied changes:** Since this comment is similar to the last general comment, we invite you to consider our response in the end of part 1.1.

Line 171 'precipitations' changed to 'precipitation'.

**Response and applied changes:** Text modified accordingly here and in the Fig.1

caption.

Line 224 APCI needs to be explained for the wide readership of the journal. What is'LGLTPE-ENS de Lyon'?

**Response and applied changes:** The APCI acronym has been deciphered. The laboratory of analysis (LGLTPE-ENS de Lyon) has been precised too.

Line 226-230 I found from the Result part (figure 4) that the authors have identified a series of 7-methyl brGDGTs, which were not widely seen in soils and lakes. Please provide the details as to how these compounds were identified and assigned in the text so that reviewers can assess whether they are identified in a right way.

Responses and changes listed following the first general comment of Referee 1.

Line 261 and Figure 3 captions What are 'AP' and 'NAP'? Please define this term prior to use.

**Response and applied changes:** AP is the acronym of Arboreal Pollen while NAP is for Non-Arboreal Pollen. They are explained on lines 290 and 291.

Line 279-280 Please show the determination coefficients and p values for the correlations Table 1 It is a little bit confusing to see so many abbreviations in the table. Please provide notes below the table.

**Response and applied changes:** The determination coefficient is already given in

the column R$^2$ selected. The MAT and WAPLS methods modeled in *Rioja* package from R give no p-value. The labels in the table have been changed following your recommendations: the spatial range of the database and the climate parameters deciphered.

Line 312 '74.6%' I think this maybe the average abundance of each compound for all the soil samples. Please specify.

**Response and applied changes:** It is the average abundance of each compound for sediment samples. To make this clearer the sentence (L. 343) has been changed in " In the MMNT5C12 sediments, isoGDGTs are dominated by $GDGT_0$ and crenarcheol (74.6% and 9.8% in relative abundances, respectively, in Fig. 4.A, grey boxplots). These compounds previously considered as lake-produced (Schouten et al., 2012) exist in moss samples (32.7% and 31.3%, green boxplots) and in soils (57.4 % and 26.7%, orange boxplots). "

Line 316 diverge from surface soil samples? Please make it clear.

**Response and applied changes:** We have changed " surface " by " soil " (L. 348).

Line 316-317 The same iGDGT distribution between soil and lake sediments does not necessarily indicates a significant contribution of GDGTs to the lake. GDGT-0 dominates over crenarchaeol in these soils, probably reflecting a high alkalinity of the soils or a dominance of methanogenic Euryarchaeota due to the anoxic environments in the moss. In contrast, the dominance of GDGT-0 over crenarchaeol in lake sediments might indicate that abundant methanogenic Euryarchaeota inhabit the anoxic lake sediments yet few Thaumarchaeota live in the lake.

**Response and applied changes:** We do agree that the recent archaeal community studies and their relation to brGDGT production could help us to mitigate our discussion of isoGDGT production origin. Following your observation and making reference to Li et al., (2018a) and Besseling et al., (2018). We changed the paragraph into (L. 350-355) " IsoGDGT patterns in lake sediments do not really diverge from soil samples which can lead to postulate that the *in-situ* production of isoGDGTs in shallow and temporary lakes like MMNT5C12 is reduced (Fig. 4.A). At least, it may show that the archaeal community both in lake and in soils is dominated by methanogenic *Euryarchaeota* more than *Thaumarchaeota* (Zheng et al., 2015; Li et al., 2018a; Besseling et al., 2018). Then, it appears (Fig. 4.A) that the isoGDGT produced in soils are dominated by crenarcheol in accordance with studies on high alkalinity of the soil (Li et al., 2018a) linked to the impact of aridity (Zheng et al., 2015). "

Line 317 'soil-produced' changed to 'produced in soils'

**Response and applied changes:** Text modified accordingly.

Line 318 '[crenarcheol]' What does the bracket indicate? Please specify in the text.

**Response and applied changes:** The bracket are used to express molecular concentration. To make it more clear we have change the sentence (L. 354) by " the crenarcheol concentration " instead of " [crenarcheol] ".

Line 319 What does 'reaction' mean?

**Response and applied changes:** " reaction " is used for " response ". Text changed accordingly.

Line 327 Reduce '22.77%' to one significant figure.

**Response and applied changes:** Change into " 22.8% "

Line 341 'methyled and cyclized' changed to 'methylated and cyclized'

Line 349 Add '-' after '5 and 6'.

Line 376 Add 'brGDGTs' after '6-methyl'.

Line 396 O2

**Response and applied changes:** Texts modified accordingly.

Line 417'react' Not properly used word.

**Response and applied changes:** Word changed in " response ".

Line 442 'affects' changed to 'affect'.

Line 502 extreme

Figure 10 captions 'foe each cores' 'for each core'.

Line 567 drier

**Response and applied changes:** Texts modified accordingly.

[revised manuscript text omitted]

---

## Referee Comment (RC4) · Anonymous Referee #1 · 4 Mar 2021

It is much better now! Well done!

---

## Editor Comment (EC1) · Nathalie Combourieu Nebout (Editor) · 4 Mar 2021

Dear Authors,

I have read carefully your responses to the reviewers. It seems that you satifactorily respond to the recommendations of reviewers.

I invite you to download the revised version of your manuscript to have carefull look on the changes you have done in the text and figures before I post a decision about your manuscript publication.

[Figure]

Waiting after seeing your revised manuscript

With my best regards

Nathalie Combourieu-Nebout

---

## Author Response (AR2)

**Editor Decision: Publish subject to minor revisions (review by editor) (08 Apr 2021)**

**Correction of the manuscript CP-2020-154 : *Climate reconstructions based on GDGT and pollen surface datasets from Mongolia and Siberia: calibrations and applicability to extremely cold-dry environments over the Late Holocene.**

Lucas Dugerdil et al.

April 2021

**Contents**

**1 List of relevant changes made in the manuscript**

We thank the reviewer 3 for his attentive reading and his accurate comments. We have followed all his recommendations for this second round of minor revisions. Mainly the correction are:

- **Wording and text modifications**

  1. The SSM calibration has been run again after removing the two pond mud samples following the recommendation of referee 3.

  2. The results and discussion still follow the same trend than the previous version of the manuscript. The only main change is the mr4 which has been preferred as best model instead of the previously mr3 model.

- **Figures updates**

  1. A supplementary figure has been added into the supplementary material: an example of the peak chromatogram integration method applied in this study.

  2. Because the calibration presented in this study has also been modified (removing the 2 pond mud samples), all the results have been changed and especially the table 2, the supplementary table S3 and the figures 7, 8 and 10. The figures have been plotted again with the new calibration.

**2 Responses to the comments of Reviewer 3 (Anonymous Referee)**

**2.1 General comments:**

After reading the rebuttal and the manuscript, I feel that most of my concerns have been addressed. However, I cannot agree with the authors that they still use lake sediments in developing the calibrations for soils. Despite similar GDGT distribution between soil and lake sediments, we cannot simply conclude that GDGTs in lakes are derived from soils, because the study area is very arid and surface runoff is limited. Besides, in situ production of GDGTs in any lakes across the globe appears to be significant.

**Response and applied changes:** The calibration method presented in this study has been compiled without the 2 pond muds samples. The results are significantly similar. In this version of the manuscript you will find the new calibration without the 2 pond muds following your recommendations. We have introduce this new calibration (L. 397) has "To guarantee the homogeneity of the calibration, the SSM has been applied on the total surface dataset excepted the two pond mud samples (even if their GDGT input seems to be validated by the BIT and IIIa/IIa indexes in Supplementary Fig. S2)"

I suggest you need to show a chromatography figure in the supplementary material which contains the separation of 5-, 6-, and 7-methyl brGDGTs.

**Response and applied changes:** We do understand the purpose of such a figure. Thus, we have added the following Rebuttal Fig. 1 in Supplementary Figure S1.

[Figure]

Rebuttal Fig. 1: Example of the peak chromatogram integration method applied in this study. This chromatogram shows the IIIa compound integration (m/z = 1050). This example is extracted from the MMNT5C12 sample.

**2.2 Specific comments:**

Besides, I found a number of typos and language issues, a part of which was shown below. I invite the authors to carefully check the manuscript.
Line 103 'on' changed to 'in'
Line 141 'for' changed to 'with'
Line 196 brings
Line 244 C46 GTGTs (GTGTs with....)
Line 339 GDGT-0
Line 340-341 Consider revising this sentence
Line 343 lead to
Line 339, 347 and 348 crenarchaeol
Line 373 the first component
Line 376 as opposed
Line 380 Add an article before 'very'.
Figure 4 captions Add '-' after '6 and 7'.
Line 374 5-methyl It should be noted that Ia is not 5-methyl brGDGTs. Please delete.
Line 385 applied to
Line 386 Delete 'the' before 'Supplementary'
Line 400 delete on
Line 409 delete of
Line 412 favor
Line 413 drive
Line 437 case community's parameter
Line 450 'archaeal' changed to 'bacterial'.
Line 457 respond
Line 544 calibrations
Line 620 appears
Line 625 'in' changed to 'on'
Line 630 delete 'from'

**Response and applied changes:** All these typos and language issues have been corrected in the final version of the manuscript as well as every mistakes in the whole manuscript following

Supplementary table 1 I found that the formula for MBT' is wrong. 5-methyl brGDGTs were missing in the formula. None of 7-methyl brGDGTs should be involved. If you calculate it using this formula. The data for MBT' might be wrong. Please check the data in the manuscript.

**Response and applied changes:** Actually it is only a mistake of typo in the published version of the formula, the formula was correct within the R script. We have changed the formula in this version.

**3 Marked-up manuscript Version**

**Notes :** The following manuscript is marked in blue to highlight the modification made for this second round minor revisions.